



# Eight years of sub-micrometre organic aerosol composition data from the boreal forest characterized using a machine-learning approach

Liine Heikkinen[1], Mikko Äijälä[1], Kaspar R. Daellenbach[1], Gang Chen[2], Olga Garmash[1], Diego Aliaga[1], Frans Graeffe[1], Meri Räty[1], Krista Luoma[1], Pasi Aalto[1], Markku Kulmala[1], Tuukka Petäjä[1], Douglas Worsnop[1,3], and Mikael Ehn[1]

[1]Institute for Atmospheric and Earth System Research /Physics, Faculty of Science, University of Helsinki, Helsinki, FI–00014, Finland
[2]Laboratory of Atmospheric Chemistry, Paul Scherrer Institute, Villigen, Switzerland
[3]Aerodyne Research Inc., Billerica, MA, USA

*Correspondence to*: Liine Heikkinen (liine.heikkinen@helsinki.fi) and Mikael Ehn (mikael.ehn@helsinki.fi)

**Abstract.**

The Station for Measuring Ecosystem Atmosphere Relations (SMEAR) II is a unique station in the world due to the wide range of long-term measurements tracking the Earth-atmosphere interface. In this study, we characterize the composition of organic aerosol (OA) at SMEAR II by quantifying its driving constituents. We utilize a multi-year data set of OA mass spectra

measured *in situ* with an Aerosol Chemical Speciation Monitor (ACSM) at the station. To our knowledge, this mass spectral time series is the longest of its kind published to date, and its detailed analysis required development of a new methodology. To this purpose, we developed an efficient and robust data analysis framework utilizing machine learning tools. These included unsupervised feature extraction and classification stages to manage and process the large amounts of data. The extensive chemometric analysis was conducted with a combination of Positive Matrix Factorization (PMF), rolling window analysis,

bootstrapping, K-Means clustering, data weighting and diagnostics based algorithmic choice-making, among others. This combination of statistical tools provided a data driven analysis methodology to achieve robust solutions with minimal subjectivity.

Following the extensive statistical analyses, we were able to divide the 2012 –2019 SMEAR II OA data (mass concentration

interquartile range (IQR): 0.7, 1.3, 2.6 µg m$^{-3}$) to three sub-categories: low-volatility oxygenated OA (LV-OOA), semi-volatile oxygenated OA (SV-OOA), and primary OA (POA). LV-OOA was the most dominant OA type (organic mass fraction IQR: 49, 62, and 73%). The seasonal cycle of LV-OOA was bimodal, with peaks both in summer and in February. We associated the wintertime LV-OOA with anthropogenic sources and assumed biogenic influence in LV-OOA formation in summer. Through a brief trajectory analysis, we estimated summertime natural LV-OOA formation of tens of ng m$^{-3}$ h$^{-1}$ over the boreal

forest. SV-OOA was the second highest contributor to OA mass (organic mass fraction IQR: 19, 31, and 43%). Due to SV-OOA's clear peak in summer, we estimate biogenic processes as the main drivers in its formation. Unlike for LV-OOA, the highest SV-OOA concentrations were detected in stable summertime nocturnal surface layers. However, also the nearby



sawmills likely played a significant role in SV-OOA production as also exemplified by previous studies at SMEAR II. POA, taken as a mix of two different OA types reported previously, hydrocarbon-like OA (HOA) and biomass burning OA (BBOA),

made up a minimal OA mass fraction (IQR: 2, 6, and 13 %). Both POA organic mass fraction and mass concentration peaked in winter. Its appearance at SMEAR II was linked to strong southerly winds. The high wind speeds probably enabled the POA transport to SMEAR II from faraway sources in a relatively fresh state. In case of slower wind speeds, POA likely evaporated or aged into oxidized organic aerosol before detection. The POA organic mass fraction was significantly lower than reported by aerosol mass spectrometer (AMS) measurements two to four years prior to the ACSM measurements. While the co-located

long-term measurements of black carbon supported the hypothesis of higher POA loadings prior to year 2012, it is also possible that ACSM was less efficiently capturing short term (POA) pollution plumes. Despite the length of the ACSM data set, we did not focus on quantifying long-term trends of POA (nor other components) due to the high sensitivity of OA composition to meteorological anomalies, the occurrence of which is likely not normally distributed over the eight year measurement period.

We hope that our successfully applied methodology encourages also other researchers possessing several-year-long time series of similar data to tackle the data analysis via similar semi- or unsupervised machine learning approaches. This way aerosol chemometric analysis procedures would be further developed into yet more streamlined and autonomous directions.

## 1 Introduction

Despite the small sizes of atmospheric aerosol particles, they play an important role in the climate system. They interfere with

solar radiation via direct absorption and scattering (direct aerosol radiative effect) and participate in cloud formation and processing thereby influencing the interactions between clouds and radiation (indirect aerosol radiative effect). In addition to the size of aerosol particles, their chemical composition plays an important role determining their direct or indirect radiative effects via composition-linked parameters such as aerosol hygroscopicity (water affinity), volatility and reflectivity.

The number concentrations of aerosol particles in the atmosphere range from a few particles per cubic centimetre to even millions, so they cannot be considered individually, but are typically divided into populations, groups or classes based on e.g. some above mentioned characteristics. Thus, the classification of aerosol particles is a necessary and critical task preceding their further understanding. Real aerosol populations are spatially mixed, overlapping and smeared in the atmosphere and their physical and chemical characteristics are for the most part not discretely distributed but continuous. Therefore, practically all

classifications of atmospheric aerosol are simplifications due to their complex interactions and change processes in the atmosphere, and any divisions between classes are to some extent arbitrary and debatable selections. Nevertheless, various statistical methods can be used to perform objective, well founded aerosol classifications, and construct aerosol models which strike a good balance between mathematical robustness, complexity (or simplicity) and usability for various purposes. In the following, some common classifications are discussed.



Organic aerosol (OA) is a major sub-micrometre aerosol constituent (Zhang et al., 2007). OA can be emitted directly as primary OA (POA) or it can form in the atmosphere via condensation or uptake of oxidized organic vapours. The latter OA fraction is termed as secondary organic aerosol (SOA). Various combustion processes are the main sources of POA. These combustion processes include for example diesel combustion in car engines, which emits hydrocarbon-like OA (HOA), or biomass burning

in forms of residential heating or wild/agricultural fires, both of which emit biomass burning OA (BBOA). The number of SOA precursors in the ambient air is immense making the linking of ambient SOA observations to SOA precursors and detailed formation processes extremely challenging.

The utilization of Positive Matrix Factorization (PMF, Sect. 4.1) on OA mass spectra recorded by Aerosol Mass Spectrometers

(AMS; Aerodyne Research Inc., MA, USA; Canagaratna et al., 2007) has linked SOA to two oxygenated organic aerosol (OOA) groups characterized by volatility: semi-volatile oxygenated OA i.e. SV-OOA, and low-volatility oxygenated OA i.e. LV-OOA. These groups are alternatively also named by their degree of oxygenation: less-oxygenated OA i.e. LO-OOA and more-oxygenated OA i.e. MO-OOA. In reality, atmospheric oxidation of aerosols is a continuum process and therefore such a division is mathematical, not clear cut and to some extent arbitrary. Due to the prominent link between OA degree of

oxygenation and volatility, the SV-OOA and LO-OOA, and the LV-OOA and MO-OOA usually describe the same OA fractions, respectively (Jimenez et al., 2009;Ng et al., 2011a). LV-OOA is typically identified by an AMS OA mass spectrum dominated by a $CO_2^+$ (at $m/Q$ 44 Th in LV-OOA mass spectrum) OA fragment (Jimenez et al., 2009;Ng et al., 2010). SV-OOA in turn typically has lower $CO_2^+$ mass fraction, but a high $C_2H_3O^+$ (at $m/Q$ 43 Th in the SV-OOA mass spectrum) fragment (Jimenez et al., 2009;Ng et al., 2010). The $CO_2^+$ fragment has been linked to various organic acids (Duplissy et al., 2011),

whereas the $C_2H_3O^+$ has been thought as a marker of non-acid oxygenates (Ng et al., 2011a). Importantly, a large amount of evidence suggests that photochemical aging of OA leads to an increasingly significant contribution of $CO_2^+$ in the OA mass spectrum (Alfarra, 2004;de Gouw et al., 2005;Aiken et al., 2008;Kleinman et al., 2008;Jimenez et al., 2009;Ng et al., 2010;Ng et al., 2011a). This indicates OA transformation to more oxygenated forms upon atmospheric aging, which ultimately yields OA of low volatility. Such OA processing (aging scheme) has shown to apply for several SOA and POA types.


While the direct POA emissions can nowadays often be quite well distinguished from SOA, perhaps due to the limitations in chemical information provided by AMS-type instruments and/or the overall similarity of SOA mass spectra regardless of the source, ambient SOA source apportionment is rarely successfully conducted. Source apportionment is also generally difficult due to complexity of atmospheric aerosol chemistry, meteorological and atmospheric transport processes and inherent

methodological (both experimental and data analytical) limitations. However, SOA formation from various precursors has been a topic of numerous laboratory studies giving insights into the most dominant ambient SOA formation pathways. Biogenic volatile organic compounds (BVOC) have shown to have a high SOA formation potential upon oxidation (Hallquist et al., 2009). Although the number of different organic species in the atmosphere is enormous ($10^4 - 10^5$) (Goldstein and Galbally,





2007), isoprene and monoterpenes clearly distinguish themselves as the most emitted biogenic VOC (Guenther et al., 2012).

While isoprene-derived SOA formation is hampered by the relatively high volatility distribution of isoprene oxidation products (Hallquist et al., 2009;Surratt et al., 2010;Shrivastava et al., 2017), monoterpenes stand out as one of the major biogenic SOA precursors, due to the production of readily condensable vapours upon oxidation (Donahue et al., 2011;Ehn et al., 2014). The boreal biome, which represents ~15% of the Earth's terrestrial area making up ~ 30% of the world's forests (Prăvălie, 2018), serves an example of a region with relatively high monoterpene emissions (Guenther et al., 2012;Rinne et al., 2009).

Measurements from the boreal forests also provide evidence of high content of naturally produced biogenic SOA (Tunved et al., 2006;Yttri et al., 2011).

The current study is targeted on the analysis of OA composition at the well-established Station for Measuring Ecosystem Atmosphere Relations (SMEAR II; Sect. 2.1) located in the monoterpene-rich boreal forest of Finland. What makes this station

unique is the large amount of long-term measurements conducted at the site. We recently reported the long-term phenomenology of sub-micrometre aerosol chemical composition seasonality at the site (Heikkinen et al., 2020). We reported a high OA mass fraction of the sub-micrometre particulate matter, ranging between 50 and 80%. The current work specifically focuses on this sub-micrometre particulate matter mass fraction with a goal to gain understanding of OA composition and its variability at SMEAR II. The data analysis includes PMF on the OA mass spectra recoded by an Aerosol Chemical Speciation

Monitor (ACSM, Sect. 2.2), but due to the near-decade long mass spectral input, handling the data retrieved via PMF analyses required also the development of new analysis tools. Inspired by previously conducted statistical analyses on OA mass spectra (Äijälä et al., 2017;Äijälä et al., 2019), we tackled the analysis problem by combining and applying various advanced statistical methods and machine learning tools. After the extensive analyses, we not only report OA composition variability at SMEAR II, but equally highlight the development of the new framework for long-term OA mass spectral analysis.

**2 Measurements**

This chapter contains a brief description of the boreal SMEAR II measurement site and the ACSM measurements conducted. For a more comprehensive measurement and station meteorology descriptions, we direct the reader to Heikkinen et al. (2020).

**2.1 Station Measuring Ecosystem Atmosphere Relations (SMEAR II)**

The measurements were conducted at the SMEAR II station described in detail previously (Hari and Kulmala, 2005;Williams

et al., 2011;Heikkinen et al., 2020). SMEAR II is well known due to the broad variety of measurements taking place at the station, tracking more than 1000 different environmental parameters within the Earth–atmosphere interface (Hari and Kulmala, 2005). The station is located in Southern Finland (61°51'N, 24°17'E, 181 m above sea level) in a ca. 60-year-old Scots pine (*Pinus sylvestris*) dominated forest. The station, recognized as a rural site, has low anthropogenic emissions, apart from two nearby sawmills situated 6–7 km to southeast from SMEAR II. In case of south-easterly winds, both monoterpene and OA





concentration are elevated at SMEAR II (Eerdekens et al., 2009;Liao et al., 2011;Äijälä et al., 2017;Heikkinen et al., 2020). The dominant source of air pollutants at SMEAR II are air masses traveling from industrialized areas in Southern Finland, St. Petersburg (Russia) and continental Europe (Patokoski et al., 2015;Riuttanen et al., 2013;Yttri et al., 2011;Tunved et al., 2006). The surrounding forest emits multiple biogenic non-methane VOCs, dominantly monoterpenes (Hakola et al., 2012;Barreira et al., 2017). Monoterpenes have been recognized to yield condensable vapours at SMEAR II (Yan et al., 2016;Rose et al.,

2018;Ehn et al., 2012) known to efficiently form SOA (Ehn et al., 2014).

## 2.2 Aerosol Chemical Speciation Monitor (ACSM)

The Aerosol Chemical Speciation Monitor (ACSM; Aerodyne Research Inc., USA), described in detail by (Ng et al., 2011c), serves as the key instrument in this study. The ACSM measurements at SMEAR II, together with the data processing techniques, are documented in detail in our earlier work (Heikkinen et al., 2020). Here, we utilize ACSM data recorded between

April 2012 and September 2019. The 2019 measurements and data preparation were performed exactly the same way as for the 2012–2018 data (Heikkinen et al., 2020).

The ACSM, which is developed following the same technology as the AMS (Canagaratna et al., 2007), samples ambient air with a flow rate of 1.4 $cm^3$ $s^{-1}$ through an aerodynamic lens having ~100% transmission of ca. 75–650 nm particles in vacuum

aerodynamic diameter ($D_{va}$), but further passes through particles up to ca. 1 µm in $D_{va}$, albeit less efficiently (Liu et al., 2007). The particles are flash vaporized at 600 °C under high vacuum and ionized with 70 eV electron impact ionization. The resulting ions and their fragments are guided to a mass analyser that is a residual gas analyser (RGA) quadrupole, which scans through different mass-to-charge ratios ($m/Q$). The particulate matter detected by the ACSM is referred to as non–refractory (NR) sub–micron particulate matter ($PM_1$). The word 'non–refractory' is attributed to the instrument limitation to detect only material

flash evaporating at 600 °C and being unable to reliably measure extremely heat–resistant chemical components such as sea salt and black carbon. The word '$PM_1$' is linked to the aerodynamic lens approximate cut-off at 1 µm.

The NR–$PM_1$ reported from ACSM measurements, is a difference (diff) between the signal of particle-laden air and signal recorded when the sampling flow passed a particle filter (filtered air). In addition to the diff measurement style, which is

measured using a chopper instead of a filter in the AMS, the lack of particle sizing and the cheaper detector model are the major differences between the AMS and the ACSM. Indeed, while the AMS utilizes multichannel plate detector (MCP) gaining high signal-to-noise (SNR) ratios, the ACSM employs a secondary electron multiplier (SEM) that provides a longer lifetime at the cost of SNR. To improve the SNR, the ACSM data utilized here was 3-hour averages instead of the original sampling resolution of 30 min.


As explained previously (Heikkinen et al., 2020), the ACSM was measuring through the roof of an air conditioned container. The inlet system contained a $PM_{2.5}$ cyclone, and a 3 Lpm overflow to avoid inlet losses. From summer 2013 onwards, a Nafion



drier was included in the sampling line, which kept the sample flow relative humidity (RH) below 30%. The instrument provides the NR-PM$_1$ chemical species' mass concentration every 30 min. The mass concentration calculations, namely the

conversion from amperes to µg m$^{-3}$ were based on ionization efficiencies, routinely calibrated using size selected ammonium sulphate and ammonium nitrate particles and a TSI Condensation Particle Counter (CPC) as a reference instrument. A final collection efficiency (CE) correction was applied based on a two-month moving median comparison with a collocated differential mobility particle sizer as the commonly used composition-based CE correction (Middlebrook et al., 2012) was not applicable due to ammonium concentration being most of the time below the detection limit. A detailed description of the CE

correction is presented previously in Heikkinen et al. (2020).

## 3 Openair and time-over-land (TOL) analyses

This chapter provides a brief description of wind and air mass trajectory analyses coupled to the analysis of OA composition at SMEAR II.

### 3.1 Openair polar plots

Openair polar plots are used in the paper to show how OA composition varied under different wind direction and speed combinations (Openair polar plots using R-based package presented by Carslaw and Ropkins (2012)). The concentration fields were calculated by binning the OA component concentration data into different wind direction and speed bins. The field was then smoothed by interpolation, which was performed between grid centres. These Openair polar plots are drawn and calculated utilizing ZeFir pollution tracker (Petit et al., 2017), which is an Igor Pro (Wavemetrics Inc., USA) graphical interface for

producing Openair polar plots (among other functionalities). The wind data used for Openair polar plots was recorded at the SMEAR II mast, above the forest canopy (16.8 to 67.2 m a.g.l.) with Thies 2D Ultrasonic anemometers. The wind roses are presented in Fig. S.1.

### 3.2 HYSPLIT trajectories and TOL

The time an air mass spent over land before reaching SMEAR II was calculated hourly using 96-hour HYSPLIT (Stein et al.,

2016) back trajectories, with arrival heights of 100 m above ground level. The meteorological input used in the model was NCEP/GDAS in 2012-2013 and NCEP/GFS in 2014-2018. The former had 1° and the latter a 0.5° horizontal resolution. Trajectories were grouped to three different source regions: clean sector, Europe-sector and Russia-sector (Fig. S.2). Europe- and Russia-sectors are considered as polluted sectors as mentioned earlier in Sec. 2.1. Similar air source region classification was utilized by Tunved et al. (2006). The grouping criteria stated that the trajectory had to spend a minimum of 90% of the

time in a sector. This means that all the trajectories grouped in the clean sector have spent minimum 90% of the time in the clean sector before arriving at SMEAR II. Time spent over islands, other than the British Isles, is not considered in the time over land (TOL) calculation.





## 4 Statistical methods

This section provides an introduction to the statistical methods utilized in this study. The application of these tools is explained
later in section 5. Here, we provide the basics of the main statistical tools utilized: Positive Matrix Factorization (PMF) and its
application in aerosol mass spectrometry as well as K-Means clustering.

### 4.1 Positive Matrix Factorization (PMF) and the Multilinear Engine (ME-2)

Positive Matrix Factorization (PMF) (Paatero and Tapper, 1993;Paatero, 1997) is a widely used algorithm in chemometrics,
which helps sorting complex measurement data into factors with altering abundances, with static factor profiles without prior
knowledge regarding the factor features. More precisely, PMF approximates the measurement data matrix ($\mathbf{X}$) as a linear
combination of these constant factor profiles ($\mathbf{F}$) and their temporal proportions ($\mathbf{G}$), both $\mathbf{F}$ and $\mathbf{G}$ containing only non-
negative elements ($g_{i,k} \geq 0$, $f_{k,j} \geq 0$). The PMF model iteratively minimizes uncertainty-weighted model residuals ($Q$) using a
least squares algorithm, directing the model solution towards combinations of $\mathbf{F}$ and $\mathbf{G}$ best describing $\mathbf{X}$. The PMF equation
in matrix notation can be written as follows:

$$\mathbf{X}_{m\times n} = \mathbf{G}_{m\times p} \cdot \mathbf{F}_{p\times n} + \mathbf{E}_{m\times n},$$

where $\mathbf{E}$ equals to the model residual matrix. If written element-wise, this equation becomes:

$$x_{i,j} = \sum_{k=1}^{p} g_{i,k} f_{k,j} + e_{i,j}, \tag{1}$$

Here, the subscript $i$ is the time column index, $j$ the variable row index, and $k$ the factor index in the PMF solution containing
$p$ factors ($p$ defined by user). The following equation for $Q$,

$$Q = \sum_{i=1}^{m} \sum_{j=1}^{n} \left( \frac{e_{i,j}}{\sigma_{i,j}} \right)^2 \tag{2}$$

can then be written as

$$Q = \sum_{i=1}^{m} \sum_{j=1}^{n} \left( \frac{x_{i,j} - \sum_{k=1}^{p} g_{i,k} f_{k,j}}{\sigma_{i,j}} \right)^2, \tag{3}$$

where $\sigma$ equals the measurement uncertainty.


Importantly, the PMF algorithm is frequently solved in robust mode, in which outliers are dynamically reweighted to prevent
the PMF model fits to be pulled towards outliers. The outliers are defined as data cells, where the ratio between the model
residual and uncertainty exceeds a user-defined threshold, $\alpha$, usually set as $\alpha = 4$ (Paatero, 1997). The $Q$ values given by the
PMF model are calculated using the robust mode.




The reliability of one modelled $Q$ minimum is not usually enough. Indeed, sometimes the PMF solutions are representative of only a local $Q$ minimum instead of the global $Q$ minimum. To avoid interpretations of a PMF solution representing a local $Q$ minimum, it is recommended to start PMF from multiple different starting points, e.g. seeds. Increasing the number of seeds, preferably together with random resampling (bootstrap) (Efron, 1979), helps mapping the stability of the PMF solution. In the

bootstrapping approach, the different PMF seeds have slightly different input matrices, which contain randomly chosen rows of the original matrix. Bootstrapping is a suitable tool for PMF statistical uncertainty evaluation, if sufficient amounts of resamples are conducted (Norris et al., 2008;Paatero et al., 2014).

Multilinear Engine (ME-2) is a popular PMF solver to reduce rotational ambiguity of PMF. One advantage of it is the

possibility to introduce known **F** rows (or **G** columns) to PMF model during model initialization (Paatero and Hopke, 2009). This approach is traditionally conducted in three ways: via techniques named chemical mass balance (CMB), $a$-value, and pulling techniques (Paatero and Hopke, 2009). In CMB (Watson et al., 1984), all of the rows in **F** (i.e. all factor profiles) are known beforehand. It can be considered as a far extreme from the traditional PMF, where none of the factor profiles is known. The $a$-value approach falls somewhere between CMB and PMF. Now, certain elements of **F** or **G** can be constrained to the

PMF, and the model output variability from the constraint is given by a scalar, $a$. $a$ can be applied to the entire **F** row (or **G** columns), or alternatively to their individual elements. The more constraints and the tighter they are ($a \rightarrow 0$), the closer the $a$-value approach is to CMB. Indeed, the case of having all $p$ rows of **F** constrained with an $a$-value of zero equals the CMB method. If pulling equations are introduced to the PMF model, PMF pulls the $f_{j,k}$ (or $g_{i,k}$ in case of **G** pulling) towards a user-defined anchor during the iterative steps.


The evaluation of the appropriate number of factors in the PMF solution ($p$) can be (for example) estimated by observing the decrease of $Q$ and the ratio between Q and the expected $Q$ ($Q_{exp.}$, which is the $Q$ normalized by the degrees of freedom of the model solution) (Paatero and Tapper, 1993). If observing the decrease of $Q/Q_{exp}$ as a function of $p$, the optimal number of factors in the solution at regions, where the drop in $Q/Q_{exp}$ with increasing number of factors stops being significant (somewhat

analogous to the elbow method utilized to evaluate the number of clusters).

### 4.1.1 PMF application in aerosol mass spectrometry

The application of PMF was first utilized with the organic aerosol data matrix, obtained via aerosol mass spectrometer (AMS) measurements in 2007 (Lanz et al., 2007), and has since then become a widely used and popular method in OA source apportionment. PMF is conducted so that **F** equals the mass spectral profiles and **G** the time series, usually in μg m$^{-3}$. A

comprehensive overview of AMS PMF studies and methodologies utilized between 2007–2011 has been given previously by (Zhang et al., 2011). (Ulbrich et al., 2009) introduced thorough AMS PMF interpretation guidelines and (Crippa et al., 2014) introduced guidelines for the ME-2 $a$-value approach. Since 2011, PMF with ME-2 has also been applied successfully to ACSM data (e.g. (Fröhlich et al., 2015;Canonaco et al., 2013;Zhang et al., 2019)).



Preparation of the PMF input (organic aerosol data matrix and a corresponding error matrix) for both AMS and ACSM data can be done with their data processing software. The preparations are based on PMF Evaluation Tool (PET) Wavemetrics Igor Pro functions (Ulbrich et al., 2009). Before initializing any PMF solver (such as the ME-2), certain preparations are often necessary for optimal modelling. The $m/Q$ with low SNR (i.e. $m/Q$ having more noise than signal) are down weighted by increasing their error. (Paatero and Hopke, 2003) suggested that $m/Q$ having SNR<0.2 should be down weighted heavily or

removed from the analysis, and $m/Q$ with 0.2<SNR<2 down weighted by a factor of 2–3. Another noisy data down weighting approach was suggested by (Visser et al., 2015), where the errors are down weighted continuously with a penalty function SNR$^{-1}$, when SNR<1. These down weightings have been done either based on the average SNR across the data set or cell-wise. Another data input modification prior to PMF initialization, should be performed regarding $CO_2^+$ ($m/Q$ 44 Th) -related variables (i.e. $m/Q$ 16–20 Th and 28 Th) because the information stored at these $m/Q$ are directly estimated from $m/Q$ 44 Th.

Such high correlation between these variables would be considered in the PMF modelling with too high importance. To avoid this, $CO_2^+$ -related variables are typically excluded or down weighted accordingly.

   PMF analysis has become easily accessible for the whole AMS/ACSM community upon the development of Igor Pro (Wavemetrics inc, USA) based user friendly PMF analysis tools, such as the Source Finder (SoFi, Paul Scherrer Institute and

Datalystica Ltd., Switzerland) (Canonaco et al., 2013) and PET (Ulbrich et al., 2009). Recently, after the launch of the commercial SoFi Pro software (Datalystica Ltd., Switzerland) (Canonaco et al., 2020) also many advanced PMF methods, became available. These methods include rolling PMF (Paatero and Tapper, 1994;Parworth et al., 2015) and PMF resampling (bootstrap).

The assumption of static factor profiles serves one of the questions of the atmospheric representativeness of the PMF output. A rolling PMF approach was suggested (Parworth et al., 2015) to account for such factor profile temporal variability. In the rolling PMF approach, a PMF run is conducted a short time window at a time (the time scale for which the static factor profile is assumed valid). This time window is shifted across the data set in even smaller time steps creating overlap between PMF windows. In practise this means choosing an $n$ day time window in an $m$ day data set ($n \ll m$), and shifting the window $q$ days

at a time ($q < n$) chronologically along the time axis, until all the $m$ days are covered.

   As the rolling PMF approach results in a large amount of PMF runs, and the amount grows even larger in case of incorporating bootstrapping (typically 100–1000 seeds per PMF window), manual investigation and conclusion-making becomes very challenging. This challenge is addressed in SoFi Pro via criteria-based selection of PMF runs (Canonaco et al., 2020).The user-

defined criteria, best describing each PMF factor (for example correlation between NO$_x$ and HOA, which both are emitted from traffic), are evaluated for each PMF run, and their scores (for example the Pearson correlation coefficient $R$ between NO$_x$ and HOA) are presented. The user can then select all the PMF runs above certain thresholds (for example $R$>0.5), or select all





of the PMF runs. Such criteria-based selection of PMF runs was first introduced by Daellenbach et al. (2017) and Visser et al. (2019). Selection and averaging all of the PMF runs without criteria-based sorting would work only in the case of having all, or all but one, factor constrained. In the case of having two or more free PMF factors, it is likely that their positions in the PMF output matrices are frequently changing, i.e. being situated in different columns in **G**. In the case of constrained PMF factors, they will always appear in their pre-designated **G** columns.

### 4.2 K-Means clustering

K-Means (Ball and Hall, 1965;MacQueen, 1967;Steinhaus, 1956;Jain, 2010) is the most popular unsupervised machine learning approach utilized in data classification. It works particularly well (computationally efficient) for large data sets with a small number of well-definable clusters ($k$). The K-Means algorithm works as follows:

1. Picking $k$ number of centroids (i.e. cluster centre points), and assigning each sample (for example a mass spectrum) to its nearest centroid based on a selected distance metric, usually the squared Euclidean distance. This step is nowadays performed following the (Arthur and Vassilvitskii, 2007) K-Means++ algorithm, proven to not only speed up the clustering process, but also significantly improve its accuracy.
2. Moving the centroids to represent the new mean of the cluster.
3. Reassigning the all the points to their closest centroids (this sometimes moves points from one cluster to another).
4. Repeating steps 2 and 3 until convergence is achieved (i.e. data points stop moving between clusters and the centroids stabilize).

The goal of the K-Means clustering algorithm is to minimize the following objective function:

$$J = \sum_{j=1}^{k} \sum_{i=1}^{n} \left\| x_i - c_j \right\|^2, \tag{4}$$

where $k$ is the number of clusters, $n$ the number of data points, $x_i$ the $i^{\text{th}}$ data point and $c_j$ the centroid of cluster $j$, and $\left\| x_i - c_j \right\|^2$ represents the Euclidean squared distance function. Hence, this makes the object function, $J$, the average squared Euclidean distance between points in the same cluster.

### 4.2.1 Silhouette score

Silhouette score (Rousseeuw, 1987) is one of the many metrics available for evaluating the number of clusters present in the data set. It is calculated both based on intra-cluster distances of data points (cohesion, $a$) and their distances to points assigned in other clusters (separation, $b$). The silhouette score for the $i^{\text{th}}$ sample can be expressed as:





$$s_i = \frac{(b_i - a_i)}{\max(a_i, b_i)} \quad . \tag{5}$$

The silhouette scores range between [-1, 1]. The scores for the $i^{\text{th}}$ sample can be interpreted as follows:


1.  $s_i = -1$ ; The sample is (likely) assigned to a wrong cluster,

2.  $s_i = 0$ ; The sample is at the decision boundary between clusters,

3.  $s_i = 1$ ; The sample is well clustered.

The silhouette score ($s_i$) is calculated for an individual sample in Eq. 5, but can also be defined for clusters ($\bar{s}$) as the average over all silhouette scores of samples belonging to the cluster, or for the entire solution (average over all samples),

yielding diagnostic information on point, cluster and solution level. (Kaufman and Rousseeuw, 2009) further suggested an average cluster silhouette $\bar{s} = 0.25$ as a lower limit for weak structure and $\bar{s} = 0.50$ as a lower limit for strong cluster structures. Strong structures indicate of a good clustering result, where the samples in the cluster are very similar to each other while being very different from the samples assigned to other clusters.

## 5 The application of PMF and K-Means in the current study

The current study focuses on conducting rolling PMF on 8 years of OA data recorded by an ACSM at the SMEAR II station. First, we performed unconstrained rolling PMF runs. We used these runs to determine the common OA factor profiles through K-means clustering. Then we utilized these factor profiles as *a priori* information to perform a relaxed CMB-like rolling PMF runs to estimate their concentration time series. This section contains a detailed description of this framework. A written overview of the method is given below and the work flow is summarized Fig. 1.

### 5.1 Rolling PMF

The initial rolling PMF was conducted using the 2012–2019 ACSM data (Fig. 2a), prepared with the ACSM data processing software, i.e. the Wavemetrics (USA) Igor Pro-based ACSM Local 1.6.0.3 toolkit, as PMF input. No down weighting, based on low SNR or relation to $CO_2^+$ was conducted with the ACSM Local software. The data matrices were imported to an Igor experiment with the SoFi Pro (6.A1) toolkit, and averaged from the initial half an hour time resolution to three-hour time

resolution in order to improve the SNR. The error propagation was accounted for during averaging. Upon the initialization of the PMF matrices, all the $CO_2^+$-related variables (i.e. *m/Q* 16, 17, 18 and 28 Th) were excluded from the analysis. Then, the errors of the noisy variables (SNR<1) were weighted cell-wise by SNR$^{-1}$.

Only the *m/Q* range of 12-100 Th was included in the rolling PMF. This mass range has been typically chosen for the ACSM

PMF analysis, and it avoids introducing the ACSM internal standard, naphthalene at *m/Q* 128, to the PMF run. *m/Q* 29, 31 and





38 Th were excluded from the analysis due to unknown interferences, likely from air and instrumental issues time to time affecting these signals, and yielding mass spectra not resembling any known aerosol type.

The rolling PMF was initialized with a constant factor number of three. The decision was made based on several (standard, i.e. rolling mechanism disabled) PMF runs, having time series lengths ranging from few months to years. Three factors were considered as an upper limit of the number of factors, as a greater number would not significantly reduce $Q/Q_{exp}$ nor produce meaningful factor profiles. This step required a subjective decision.

The rolling window width was set to 30 days with 10 days window shifts. Previous studies conducted by Parworth et al. (2015) and Canonaco et al. (2020) set the window width to two weeks and the shift to one day, which is much shorter than selected here. However, as shown by Canonaco et al. (2020) the PMF solutions were seemingly equally good for window widths higher than two weeks (tests up to window width of 28 hours). Only widow widths shorter than two weeks led to a less good PMF result.. As the time span of our data was nearly eight times greater than utilized in the previous studies, we speeded up the PMF modelling process by choosing a longer window width and shift. More testing could be conducted on appropriate lengths. However, if the number of PMF runs were to increase significantly from the amount performed here, it would be feasible to perform the PMF modelling on a server. With the current settings, the rolling PMF run performed in this study using a PC, lasted 48 hours.

Finally, also, the bootstrap mechanism (resampling) was enabled, and a hundred iterations were conducted at each window. A subset of the rolling PMF input is visualized in Fig. 2b. The rolling PMF yielded 62700 factor profiles (20 900 three factor solutions) and time series, respectively, distributed in 209 PMF windows.



**1. Rolling PMF**

- 1.1. Import PMF matrices to Source Finder (SoFi)
- 1.2. Pre-process PMF input
- 1.3. Determine an upper limit of number of factors for rolling PMF
- 1.4. Set number of iterations
- 1.5. Set PMF window width and shift
- 1.6. Run rolling PMF

**2. Solutions for rolling windows**

- 2.1. Import rolling PMF result files (window-by-window) to MATLAB
- 2.2. Perform mass scaling to all mass spectra in window
- 2.3. Perform K-Means clustering with different number of clusters and calculate silhouette score for each solution
- 2.4. Select the K-Means clustering solution with the highest silhouette score
- 2.5. Calculate silhouette weighted cluster centroids for the solution
- 2.6. Combine all silhouette weighted centroids from all PMF windows in one matrix
- 2.7. Perform steps 2.1-2.6 one rolling window at the time until cluster centroids are calculated and saved from all windows

**3. Overall classification**

- 3.1. Determine the optimal number of clusters ($k$) in the matrix built in step 2.6 using silhouette scores
- 3.2. Run K-Means clustering with $k$ clusters
- 3.3. Calculate silhouette weighted cluster centroid interquartile ranges (IQR) for all $k$ clusters

**4. Rolling rCMB**

- 4.1. Import PMF matrices to SoFi
- 4.2. Pre-process PMF input
- 4.3. Upload the $k$ cluster centroid IQRs
- 4.4. Set the number of factors to $k$
- 4.5. Constrain the factors with the $k$ cluster centroid IQRs
- 4.6. Initialize rolling rCMB otherwise similarly as the initial rolling PMF
- 4.7. Run rolling rCMB
- 4.8. Average over all rolling rCMB windows
- 4.9. Evaluate the solution reliability
- 4.10. Explore final results

**Figure 1** Work flow describing the machine learning analysis approach utilized in the current study. The method comprises four main phases: 1. Performing rolling PMF, 2. Performing window-by-window (file-by-file) clustering of rolling window iterations (Phase I clustering), 3. Conducting overall classification of the centroids calculated for all PMF windows (Phase II clustering), and finally 4. Performing rolling relaxed- chemical mass balancing using the centroids retrieved in the previous step as CMB anchors. Sections 4 and 5 in the paper introduce all the vocabulary needed for understanding this figure. These sections also contain detailed descriptions of each step in the method.




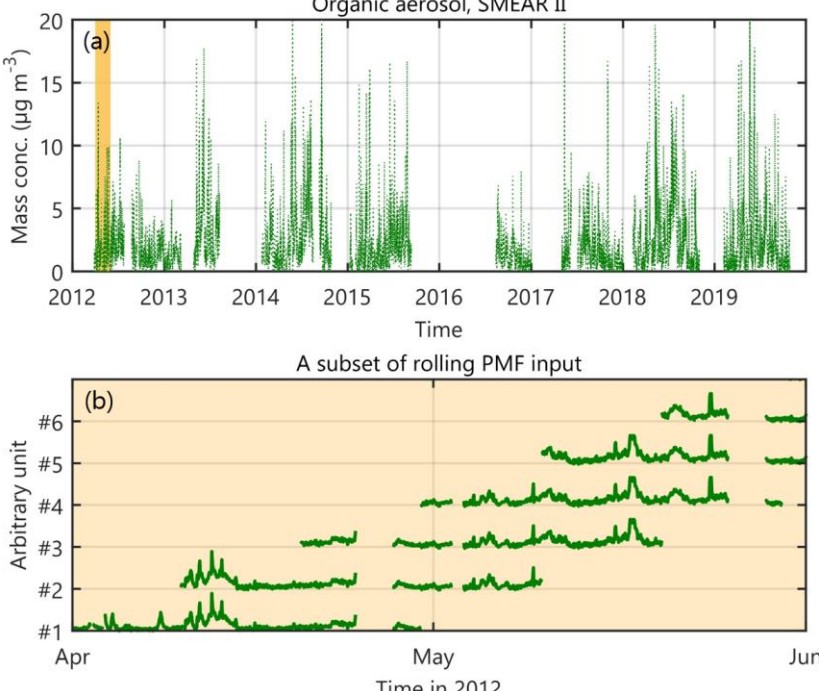

**Figure 2** (a) The time series of OA measured at SMEAR II and utilized in the current study. The y-axis represents OA mass concentration in µg m$^{-3}$ and the x-axis the time. The figure also depicts the data coverage within the eight years. The yellow shaded region represents the first two months of measurement data, which are further shown in panel b. (b) Schematic figure visualizing the rolling window approach.
Now, *x*-axis spans from April 1$^{st}$ to June 1$^{st}$, 2012 and the six OA time series represent the timespans of successive rolling PMF windows. With the settings used in the current study, this two-month period would be part of six rolling PMF windows.

## 5.2 Mass spectra scaling and K-Means clustering

Sorting the rolling PMF output via criteria-based selection/sorting into three factors would have required a significant understanding of the PMF output beforehand. Selecting solid criteria can be straightforward near known pollution sources, but 375 in case of multiple unknown factors and distant sources such becomes complicated. SMEAR II represents a station with minimal anthropogenic sources. To exemplify the challenges in correlation-based criteria at SMEAR II, we can take the correlation between NO$_x$ and HOA as an example. Both of these species are emitted from traffic and known to correlate well near traffic sources. However, in the case of transported traffic emissions, many things can affect the life time of the emitted species, which affects the correlation between the emissions at SMEAR II. If we pick the effect of wet deposition as an 380 example, it will remove the particulate HOA much more efficiently than gaseous NO$_x$. This complexity served as one of the main drivers for using mass spectral clustering in this work to explore the presence of POA rather than attempting to understand the POA correlations with external data (such as NO$_x$). Therefore, we approached the sorting challenge via clustering (K-





Means algorithm) the rolling PMF output mass spectra. The clustering-based sorting was conducted PMF window-by-window (Phase I; See detailed description in Sec. 5.2.1). This step was followed by exploring the number of clusters across all PMF

windows by further clustering all the Phase I cluster centroids (Phase II; See detailed description in Sec. 5.2.2). All the clustering procedures conducted in this study were performed within MATLAB 2017a using the *kmeans* algorithm, which utilizes K-Means++.

### 5.2.1 Solutions for rolling windows (K-Means clustering Phase I)

The rolling PMF output was uploaded into MATLAB from Hierarchical Data Format (HDF) -files created for each PMF

window, respectively, during the ME-2 modelling process. Prior to clustering, we scaled the PMF output with the following function suggested by (Stein and Scott, 1994):

$$\text{weight}_{\frac{m}{Q}} = \left(\frac{m}{Q}\right)^{s_m}, \tag{6}$$

where $m/Q$ equals the mass-to-charge ratio ranging from 12–100 Th, and $s_m = 1.36$ (recommendation by Äijälä et al., 2017). Each signal at each $m/Q$ was multiplied by its $m/Q$-corresponding weight-value. As recommended by Äijälä et al., 2017, the usage of this scaling factor gives gradually more weight to the patterns at the end of mass spectrum, containing a lot of

information regarding OA sources.

Importantly, the following clustering of bootstrap iterations one rolling window at a time was conducted using cosine (dis)similarity (Sokal and Sneath, 1963) as the K-Means distance metric. The cosine similarity metric has been popular in mass spectral comparisons (Stein and Scott, 1994). It describes the similarity between two $n$-dimensional ($n$, i.e. the number of $m/Q$

was 70 in our study) vectors (**A** and **B** in the equation below) via the cosine of the angle between them. Hence, the metric is not magnitude but orientation dependent. In our case this also meant that normalization of the weighted mass spectra was not necessary. The cosine (dis)similarity is defined as follows:

$$\text{Cosine (dis)similarity} = \frac{\mathbf{A} \cdot \mathbf{B}}{\|\mathbf{A}\|\|\mathbf{B}\|}, \tag{7}$$

where **A** and **B** are n-dimensional vectors, which in the current case would correspond to two mass spectra.

Finally, the PMF window-by-window clustering of bootstrap iterations was conducted as follows:
1. Clustering (MATLAB 2017a *kmeans* function using cosine (dis)similarity as the distance metric) and calculating mean silhouette values (MATLAB 2017a *silhouette* function using cosine (dis)similarity as the distance metric) for 2–4 clusters per PMF window. This step was performed using the 300 mass scaled (Eq. 6) mass spectral profiles (3 factor profiles, 100 iterations) given by the 30-day rolling window.





2. Finding the number of clusters achieving the highest mean silhouette value in the PMF window. Only this clustering result was used in the following steps as it was considered as the "best solution".

3. Un-doing the mass scaling and calculating silhouette-weighted cluster centroids (here: the median of all mass spectra belonging to the cluster, each multiplied by their spectra-specific silhouettes) for each PMF window. The weighting of the cluster centroid calculation by silhouette scores was performed similarly to Äijälä et al. (2017, 2019) studies: all mass spectra possessing a negative silhouette score were discarded from the cluster centroid calculation and the rest of the mass spectra were multiplied by their spectra-corresponding silhouettes. This way, the spectra with the highest silhouette scores would influence the cluster centroid the most, and the spectra with the lowest silhouette score were either discarded (if silhouette score is zero or negative) or having minimal weight on the final cluster centroid. This step helps to alleviate possible K-Means susceptibility to outliers in clusters.

4. Appending the silhouette-weighted cluster centroids in a matrix ($F_I$). If the PMF window was clustered with three factors in the 3rd step listed here, then $F_I$ would gain three new rows: one for each cluster centroid mass spectrum.

5. Moving to the next PMF window and repeating steps 1-6 until all PMF window are clustered and matrix $F_I$ contains all the silhouette-weighted centroids from each PMF window.

All the steps presented above, were done programmatically in MATLAB. The final number of mass spectra stored in $F_I$ was 479. The overall mean silhouette values for 2-4 clusters were high, strongly indicating segregation of strong cluster structures in the PMF window-by-window clustering of bootstrap iterations (Fig. 3a). The optimal number of clusters in the PMF windows was 2 in ca. 80% of the PMF windows (Fig. 3b), which meant that only ca. 20% of the PMF windows contained 3–4 different resolvable PMF factors.



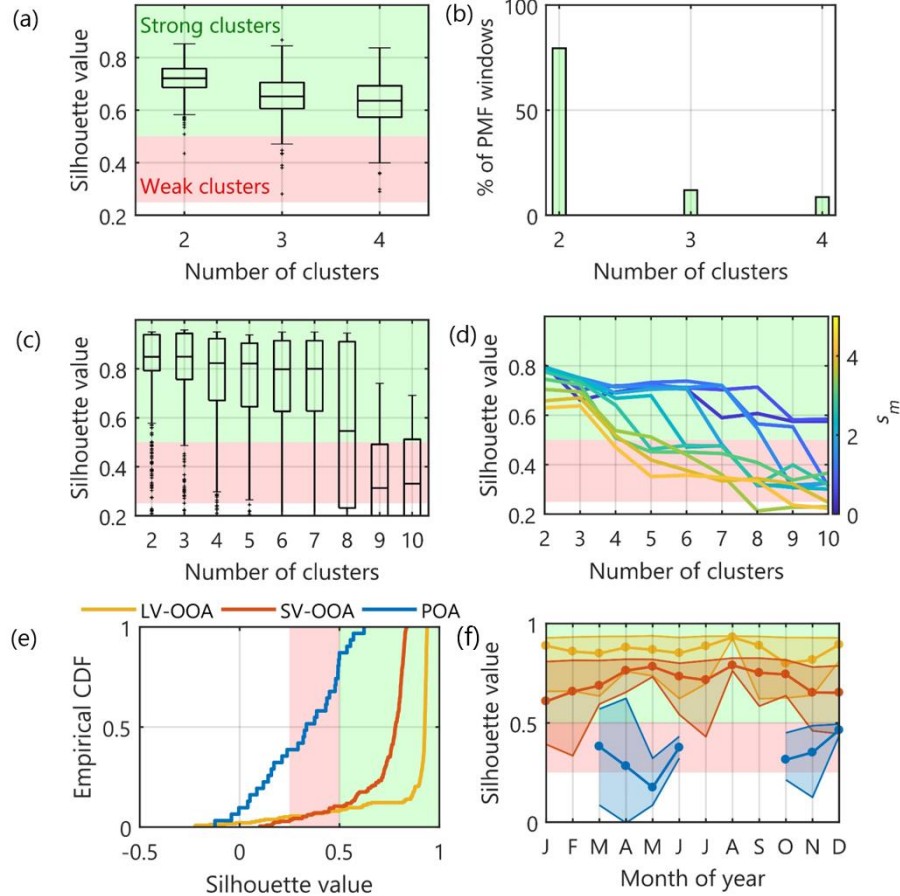

**Figure 3** (a) Box-and-whisker diagram displaying the silhouette score distribution for $k$ (number of factors) = [2, 4] representing all 209 PMF windows (Phase I). The green and red shadings indicate the ranges of strong and weak cluster structures, respectively. (b) Fraction of PMF windows achieving the highest silhouette score when the number of clusters ($k$) was 2, 3 or 4. (c) Silhouette score distribution for $k$ = [2, 10] for Phase II (i.e. clustering the 479 profiles obtained from the 209 PMF windows in Phase I). (d) Evolution of the median silhouettes in $k$-space as a function of the mass scaling (Eq. 6) factor, $s_m$, which gives dynamically more weight to the end of the mass spectrum. The colour scale presents the $s_m$ value for each line. (e) Cumulative distribution function (CDF) of the $k$ = 3 Phase II silhouette scores for the three clusters (named LV-OOA, SV-OOA and POA), respectively. This subplot shows that POA has the weakest cluster structure, and LV-OOA the strongest. (f) Temporal behaviour of the median silhouette score of each cluster in the $k$ = 3 Phase II solution. Here, each month displayed must contain a minimum of 30 days of cluster appearance, explaining the gap in the POA seasonal cycle, as it is not as frequently resolved as the other clusters.





### 5.2.2 Overall classification of mass spectra (K-Means clustering Phase II)

The next step was to explore the dominant mass spectral clusters in the whole data set. Phase II contained the following steps:

1.   Performing mass scaling (Eq. 6) for $F_I$ mass spectra, as performed earlier in the PMF window-by-window clustering of bootstrap iterations (Phase I; Sec. 5.2.1).

2.   Calculating mean silhouette scores (MATLAB 2017a *silhouette* function using cosine (dis)similarity as the distance metric) for 2–10 clusters.

3.   Exploring how many clusters are needed to gain the highest mean silhouette score. In case of a vague difference
450         between silhouettes (as shown in Fig. 3c), the step is followed by performing steps 1 and 2 again with different mass scaling $s_m$-values. The optimal number of clusters should preserve the high silhouette score even at high $s_m$-values. We explored $k = [3, 6]$ solution space with different $s_m$-values ($s_m = [0, 5]$). By increasing $s_m$, the silhouette value for $k = 3$ increased to the same level as $k = 2$, while $k > 3$ solution silhouettes decreased below the strong cluster limit (Fig. 3d). We thus selected three clusters for the following steps.

4.   Clustering (MATLAB 2017a *kmeans* function using cosine (dis)similarity as the distance metric) the mass weighted mass spectra ($s_m = 1.36$) with the number of clusters defined in the previous step.

5.   Un-doing the mass scaling, and calculating silhouette-weighted, normalized cluster centroids (cluster median) and the cluster mass spectral variability (lower and higher quartiles). These cluster centroids represent the prevailing OA types in SMEAR II sub-micrometre aerosol.

The three different OA clusters found by this method were named low-volatility oxygenated organic aerosol (LV-OOA), semi-volatile oxygenated organic aerosol (SV-OOA) and primary organic aerosol (POA). The LV-OOA and SV-OOA clusters had generally high silhouette scores whereas the POA cluster had a weaker structure (Fig. 3e). More discussion on the mass spectral features is provided in the results section (Sect. 6.1).

### 5.3 Rolling relaxed-CMB (rCMB)

After gaining the prevailing OA types mass spectral features via the above explained clustering processes, we wanted to gain understanding of the temporal features and mass loading of each OA type. As the HDF-files for each rolling PMF window also contain time series information for each factor profile, we were able to calculate cluster-specific time series utilizing these time series connected to each cluster member spectra. The time series of the OA types were discontinuous since factors were not resolved in every window. Therefore, we utilized the silhouette-weighted cluster interquartile ranges (IQRs) gained in
Sect. 5.2.2. to constrain a rolling relaxed CMB (rCMB) run to gain continuous time series for each OA type. These cluster-specific time series extracted from the initial PMF were afterwards used to evaluate the rCMB run (Sect. 5.3.1), but also enabled us to explore the silhouette score temporal behaviour. The silhouette score monthly medians are visualized in Fig. 3f.



Only SV-OOA showed some seasonality. Due to the stability in the monthly median silhouettes, we consider the mass spectral classification robust.


The rCMB run was conducted via rolling PMF using the cluster centroids of the OA factor profiles as *a priori* information. After extracting the governing mass spectral features across the data set, we exported the silhouette weighted and normalized mass spectra to SoFi Pro 6B. We set up a PMF run with three factors, all of them constrained with our silhouette-weighted cluster centroids (median factor profiles). However, differing from the traditional CMB approach, we passed ME-2 the allowed

limits within which the factor profiles should vary. These limits were the 25th percentile (lower limit) and 75th percentile (higher limit) of the silhouette-weighted cluster centroid spectra. The rCMB was otherwise initialized exactly like the initial rolling PMF run. The $CO_2^+$ related variables were excluded, and the errors of the weak variables were treated similarly (cell-wise $SNR^{-1}$ penalty function). The rolling window length was again 30 days with a 10 day shift, and resampling was enabled with 100 seeds. $m/Q$ 29, 31 and 38 Th were still discarded from the analysis. The final rCMB results for each factor, respectively, were

obtained by averaging over the 20 900 PMF runs for each time point (in total: $3 \times 20\ 900 = 62\ 700$ factor profiles and time series). As all the factor positions in rCMB were fixed (LV-OOA profile was constrained at the **F** matrix first row, SV-OOA at the second and POA at the third), such averaging was appropriate.

### 5.3.1 rCMB output evaluation

To evaluate the averaged rCMB output, we first compared the $Q/Q_{exp}$ values between the initial rolling PMF and rolling rCMB.

The comparison of the $Q/Q_{exp}$ retrieved from each iteration in each rolling window is visualized in Fig. 4a. As expected, the mean rCMB $Q/Q_{exp}$ value is higher (38% increase) than that of the initial rolling PMF $Q/Q_{exp}$. To continue the rCMB result evaluation via residuals, we investigated rCMB model uncertainty-scaled residuals (**R** matrix, $r_{i,j}$ in cell notation in Eq. (8)). **R** elements were calculated with SoFi Pro using the following equation:

$$r_{i,j} = \frac{e_{i,j}}{\sigma_{i,j}}, \tag{8}$$

where $\sigma_{ij}$ indicates the measurement error provided in the initial PMF input error matrix and $e_{ij}$ the model residual (i.e. the

difference between model input and model output: $x_{ij}$ (meas.) – $x_{ij}$ (mod.)). A normalized scaled residual histogram is presented in Fig. 4b. The scaled residual histogram is fairly unimodal and spreads between [-4, 4] (most data between [-3, 3]) as desired (Paatero and Hopke, 2003).

The scaled residual mass spectrum (Fig. S.3a) showed that the spectrum IQR scattered mostly between [-1, 1], but medians

were often slightly negative ($-0.05 > r > -0.5$) implying minor rCMB overestimation of those $m/Q$. To explore which of the rCMB factors would be most influenced by overestimations, we calculated cosine (dis)similarity angles (Eq. 7) between the residual mass spectrum and the rCMB factor profile mass spectra (Fig. 5, discussed in Sec. 6.1). The cosine similarity values



(Eq. 7) were -0.01, -0.1 and -0.4 for LV-OOA, SV-OOA and POA respectively. As the cosine similarity absolute value was highest for POA, it is possible that POA was slightly overestimated by rCMB. These (possibly POA) overestimations seemed

to consistently occur in summer ($r \sim$ -0.2) as displayed in Fig. S3b. Still, these values of rCMB summertime overestimations were small and well within the desired limits ($r = $ [-3, 3]; Paatero and Hopke, 2003). However systematic over- or underestimations are obviously not desired. We suspect that this summertime rCMB overestimation was yielded from the limits given to ME-2, which were the same throughout the data set (overall cluster silhouette-weighted IQR). With the method presented here, we could easily extract time-dependent limits for ME-2 variability. However, introducing such to dynamic

approach to ME-2/SoFi Pro is not yet possible. Importantly, no long-term trends were observed in the scaled residual behaviour (Fig. S3c). Finally, the conclusion to be drawn from this residual analysis is that the rCMB result captured the data variability well, but time-dependent limits for reference profile variability in rolling rCMB could likely further improve the result and possibly reduce the minor (POA) summertime rCMB overestimation.

The comparisons between rCMB time series (Fig. S.4) to the cluster-specific time series serve as the final step in rCMB validation. The overall Pearson correlation coefficient between the mean-cluster time series and the sum of rCMB factor time series is approaching unity ($R = 0.99$), and the correlations between different OA classes are 0.97, 0.94 and 0.78 for LV-OOA, SV-OOA and POA, respectively (Fig. S.4). In fact, such high degree of agreement indicates very good rCMB performance in retrieving time series for the different OA classes. As a final note, as discussed previously, the POA appearance in the time

series retrieved after the Phase II clustering was likely depending on the POA mass fraction in different PMF windows. We evaluated that 95% ($3\sigma$) of the PMF windows where POA was not classified, had a POA mass fraction (i.e. the mass fraction of POA in relation to the total rCMB OA mass; $f_{POA}$) of 6% (Fig. S.5a), when POA explained variation (i.e. rCMB-derived variability explained by POA compared to the total measurement variability) was 7% (Fig. S.5b). Such numbers resemble the PMF "rule-of-thumb" detection limit of ca. 5% estimated by Ulbrich et al. (2009). This final note indicates simply that the

POA cluster was not found when the POA concentration was near-zero in rCMB. Such behaviour is certainly a factor explaining the slopes between the cluster-specific time series and rCMB time series presented in Fig. S.4.





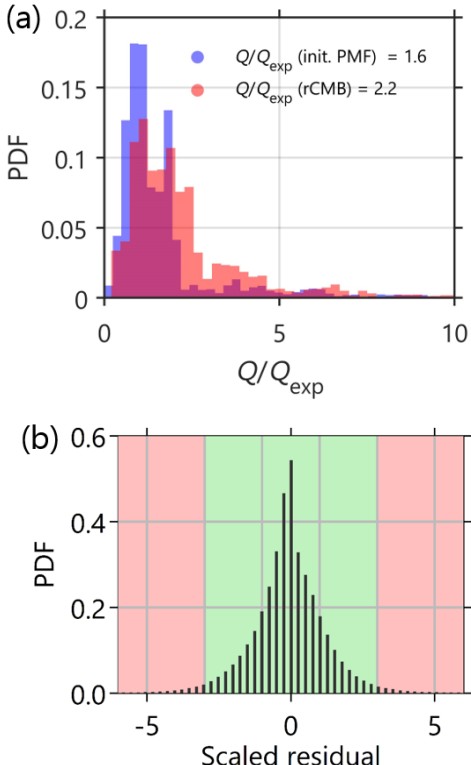

**Figure 4** (a) Normalized histograms (probability density function, PDF) of the initial rolling PMF run (blue) and the rolling rCMB run (red).
The legend reports the mean $Q/Q_{exp}$ values of both experiments. (b) The PDF of scaled residual (uncertainty-weighted residual) in the rCMB run. The green shaded area represents the area within which the scaled residual is desired to be distributed in (Paatero and Hopke, 2003). The red shaded areas represent the non-desired regions.

## 6 Results and discussion

In this section, we introduce the key features of the LV-OOA, SV-OOA and POA clusters' mass spectra (Sec. 6.1). After the

detailed mass spectral investigation, which explains the naming of each cluster, we further discuss the temporal behaviour of

these OA classes (data retrieved via rCMB; Sec. 6.2). The section then includes a brief analysis of wind direction and speed

dependences of the OA classes (Sec. 6.3.1) via Openair polar plots (Carslaw and Ropkins, 2012; Petit et al., 2017). As a final

section in this chapter we explore LV-OOA, SV-OOA and POA loading as a function of time over land in the clean sector

(Sec. 6.3.2) to yield understanding on natural OOA production over the NW Nordic quadrant of SMEAR II.

### 6.1 Mass spectral features of OA clusters

The cluster centroids resulting from the overall classification of SMEAR II mass spectra serve as one of the key results of the

current study (Fig. 5). The three OA classes were named already previously as low-volatility oxygenated organic aerosol (LV-





OOA), semi-volatile oxygenated organic aerosol (SV-OOA) and primary organic aerosol (POA), but we start this chapter by motivating the decisions behind each OA cluster name.


The naming of LV-OOA was based on the dominance of $m/Q$ 44 Th in the mass spectrum, and the naming of SV-OOA was done due to the high $m/Q$ 43 Th (higher than $m/Q$ 44 Th). The naming of the POA was motivated based on the resemblance of the POA mass spectrum with both hydrocarbon-like OA (HOA) and biomass-burning OA (BBOA). The cosine (dis)similarities between POA and HOA or BBOA (both references from (Ng et al., 2011b); spectra downloaded from

http://cires1.colorado.edu/jimenez-group/AMSsd/, last access June 3rd, 2020; Ulbrich et al., 2009) were 0.85 and 0.80, respectively. If a mass scaling (Eq. 6 with various $s_m$) was applied to all spectra, the cosine (dis)similarities between POA and HOA and BBOA, respectively, fast exceeded 0.90. This possibly happened, because less weight was given to $m/Q$ 44 (and 43 Th), which is higher in our POA than in typical fresh HOA or BBOA spectra (see for example Ng et al., 2011b) likely meaning that our POA cluster is more oxidized than fresh POA. As we expect HOA and BBOA to be primary in origin, and our cluster

centroid spectrum resembles both of them, we decided to call this OA class POA.

To further motivate our selection of names for the three clusters (as well as to visualize the cluster structures for the readers), we displayed all the different mass spectra belonging to each cluster in an $m/Q$ 43 Th vs $m/Q$ 44 Th organic signal contribution space ($f_{44}$ vs $f_{43}$ space; Fig. 6a). Ng et al. (2010) first introduced this projection, also called the 'triangle plot'. This perspective

separates well the LV-OOA, SV-OOA and POA clusters. They are placed in each corner of the triangle in Fig. 6a. LV-OOA lies on the top of the triangle, exhibiting the highest OA mass fraction of $m/Q$ 44 Th (i.e. $f_{44}$; hereafter this same nomenclature logic is used also for other OA mass fractions of various different $m/Q$), whereas SV-OOA and POA lie at the bottom of the graph possessing nearly equally low $f_{44}$. The $f_{43}$ on the other hand, is highest for SV-OOA, and lowest for POA (nearly equally low as for LV-OOA).


By using a parametrization provided by (Canagaratna et al., 2015), we converted the $f_{44}$ vs $f_{43}$ plot into a hydrogen-to-carbon ratio ($\mathrm{H:C} = 1.12 + 6.74 \times f_{43} - 17.77 \times f_{43}^2$) vs oxygen-to-carbon ratio ($\mathrm{O:C} = 0.079 + 4.31 \times f_{44}$) space (Van Krevelen (VK) diagram (Van Krevelen, 1950); Fig. 6b). The bulk OA data from AMS measurements has been shown to follow a -1 slope on the VK diagram (Heald et al., 2010), where the most fresh OA has the highest H:C and lowest O:C and the aged OA

the opposite. The evolution of OA in the VK space following different lines results mainly from OA functionalization. In case of a slope of 0, OA functionalization would occur mostly by addition of alcohol or peroxide groups. In case of a slope of -1, carboxylic acid groups are being added and the slope of -2 would indicate additions of ketone or aldehyde groups. Factorized OA data were previously visualized in the VK diagram by (Ng et al., 2011a), where the slope for OOA data was ca. -0.5. They suggested that ambient OOA aging would result from addition of alcohol and peroxide functional groups without introducing

fragmentation and/or the addition of carboxylic acid groups with fragmentation. Here, we visualize only SV-OOA and LV-OOA, as they provide better statistics than POA as number of objects in POA cluster was small, and these points would be



highly scattered in the VK diagram. Furthermore, it is also mentioned in Canagaratna et al. (2015) that the parametrization works less well for POA.

Before interpretation of the VK diagram, we revisit results from European ACSM intercomparisons conducted at Aerosol Chemical Monitor Calibration Center (ACMCC). A large variability within $f_{44}$ was observed between different ACSM units (Crenn et al., 2015;Fröhlich et al., 2015;Freney et al., 2019). Furthermore, the observed $f_{44}$ were systematically higher than the $f_{44}$ measured with a co-located high resolution AMS, which was shown to give consistent O:C for a suite of organic samples with known O:Cs. While the $f_{44}$ variability was not significantly propagated in OA class mass fractions retrieved with PMF
analyses of co-located ACSM data sets (Fröhlich et al., 2015), the O:C ratios of different classes were naturally affected (as O:C parametrization for AMS-type instruments is directly $f_{44}$ dependent). The $f_{44}$ variability has been to some extent explained by an AMS/ACSM vaporizer artefact, which leads to a release of $CO_2^+$ in the presence of high nitrate mass fractions (Pieber et al., 2016;Freney et al., 2019). Even though the presence of $m/Q$ 44 Th has been minor in our ammonium nitrate calibrations, and the nitrate mass fraction is generally low at SMEAR II, we cannot be sure whether the $f_{44}$ and thus the O:C-ratios presented
in the VK are overestimated. Thus, the absolute O:C-values should be interpreted with caution. However, if comparing the VK diagram to the VK diagram drawn by (Ng et al., 2011a) representing from 43 ambient AMS datasets, we can see that the our SV-OOA O:C is similar to the SV-OOA O:C retrieved by Ng et al. (2010), but our O:C for LV-OOA is higher. Still, our LV-OOA values do resemble those retrieved by Äijälä et al. (2019) with an AMS.

In general, the separation of SV-OOA and LV-OOA in the VK is distinct: the O:C of SV-OOA is ca 30% of the LV-OOA O:C. The SV-OOA H:C is highest, and stays rather constant in the SV-OOA cluster data cloud (slope = 0, slope of adding alcohol or peroxide groups), whereas the H:C decreases as a function of O:C in the LV-OOA cluster data cloud. Due to the scatter in the LV-OOA data cloud we do not aim on quantifying a slope for it.

The second row of projections visualized in Fig. 6 focuses on visualizing key POA characteristics. The $f_{44}$ vs $f_{60}$ visualization used in Fig. 6c is common to distinguish fresh BBOA from aged OA (Cubison et al., 2011). The lower the $f_{44}$ is, the more fresh the OA is expected to be, and the higher the $f_{60}$ is, the higher the fresh BBOA fraction. The POA captured most of the high $f_{60}$ cases (i.e. cases with $f_{60}$ above determined background of 0.003 (Cubison et al., 2011)), and the rest (which also had the highest $f_{44}$) were included in the LV-OOA cluster. These were clear LV-OOA cluster outliers as these spectra silhouette scores were
all below 0.20. If moving to Fig. 6d, i.e. an $f_{55}×f_{57}$ vs $f_{60}$ diagram, we can see that these high $f_{60}$-containing LV-OOA points are situated at the bottom of the plot, and all POA objects score a much higher $f_{55}×f_{57}$. $f_{57}$ has been associated with HOA (Zhang et al., 2005), while $f_{55}$ is present in HOA mass spectra usually at equally high contributions. However, $f_{55}$ is not a good HOA marker alone, as it is present in all of the mass spectra (Fig. 5). Thus, the y-axis in Fig. 6d was chosen to be a product of the two instead of a sum of the two, as in this way a high $f_{55}$ (often the case with biogenic SOA) with marginal $f_{57}$ would not be
classified as a HOA marker. To conclude, Figs. 6c&d visualize how POA contains both HOA and BBOA features.



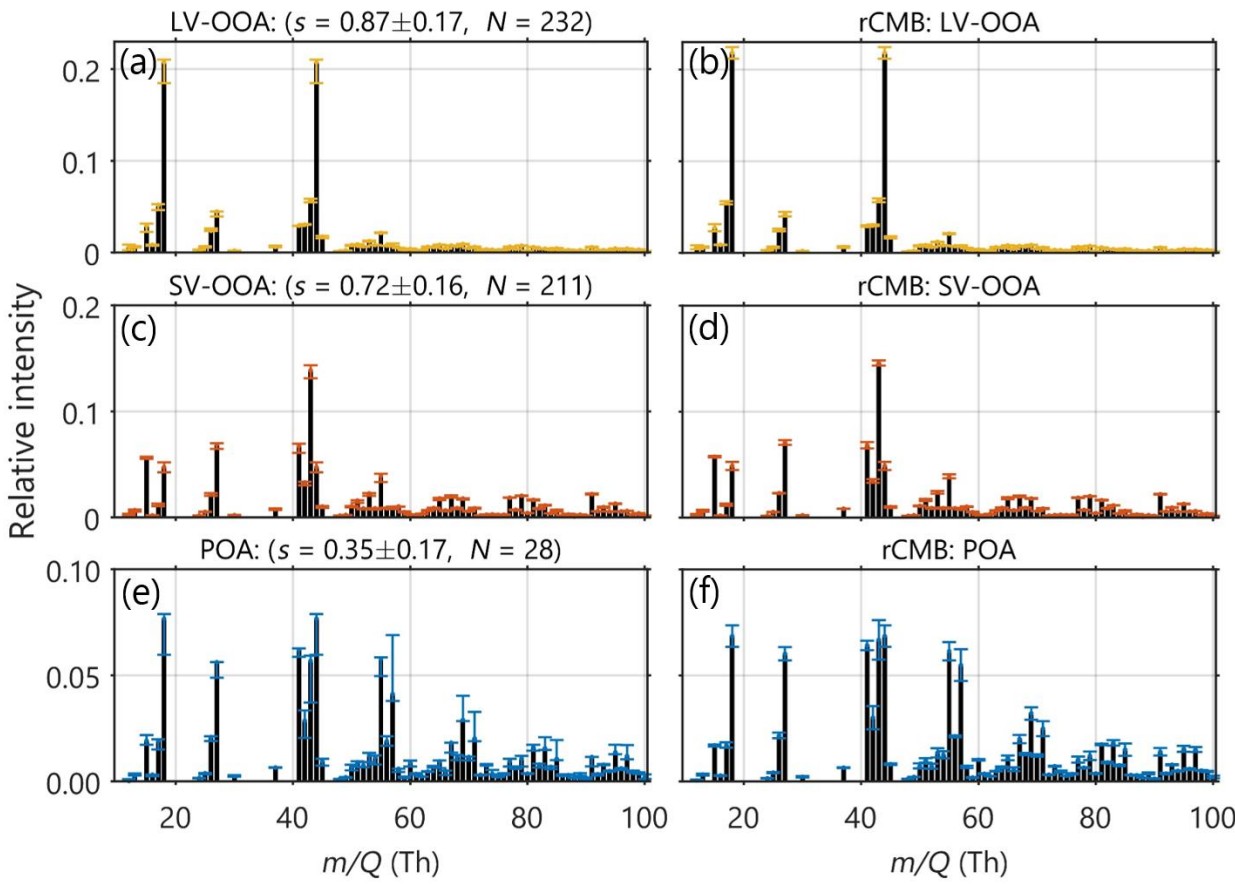

**Figure 5** The left panels (a, c, e) represent silhouette-weighted median cluster centroid mass spectra obtained when the number of clusters (*k*) equals 3 in Phase II K-Means clustering (final result). Here, *y*-axis indicates the relative signal intensity and *x*-axis the mass-to-charge ratio (*m/Q*) the cluster centroid mass spectra identified as low volatility oxygenated organic aerosol (LV-OOA), semi-volatile oxygenated organic aerosol (SV-OOA) and primary OA (POA). The panel titles include the mean ± standard deviation of the cluster silhouette score (*s*), and the number of spectra belonging to each cluster (*N*). The error bars visualize the 25th and 75th percentiles (i.e. the lower and higher quartiles). The right panels (b, d, f) show the mean LV-OOA, SV-OOA and POA mass spectra obtained from rCMB. The error bars visualize the standard deviation of each *m/Q* signal fraction.




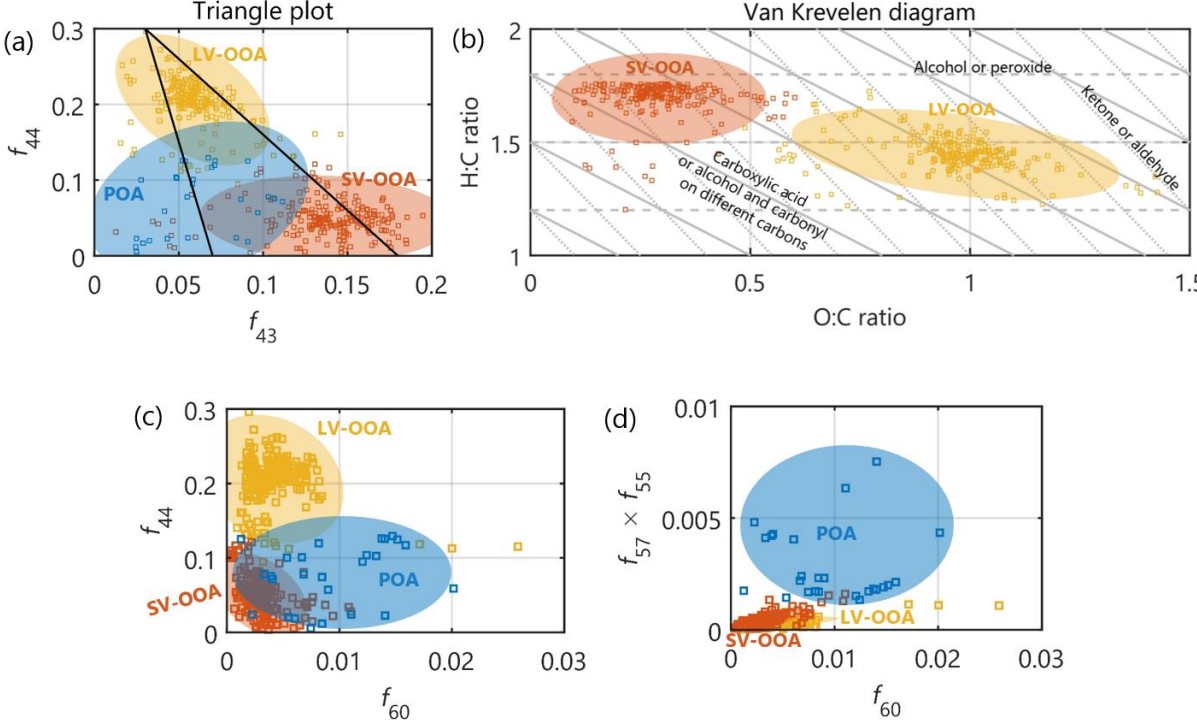


**Figure 6** (a) A triangle plot visualizing the mass spectra distribution in each cluster in $f_{44}$ vs $f_{43}$ space, (b) Van Krevelen diagram visualizing the mass spectra in H:C vs O:C space for LV-OOA and SV-OOA, (c) mass spectra in $f_{44}$ vs $f_{60}$ space for indications of fresh BBOA, (d) $f_{55}$ × $f_{57}$ vs $f_{60}$ space for indications of HOA and BBOA. The ellipsoids in panels (a) and (b) and LV-OOA and SV-OOA ellipsoids in panels (c) and (d) are calculated to contain 95% of the data (assuming normal distribution). The POA shadings in panels (c) and (d) are drawn by hand
to guide the eye due to strong log-normal distributions of the data displayed.

## 6.2 Temporal variability of OA composition

This section contains the analysis of the OA components' time series retrieved via rCMB. These time series are visualized in in monthly resolution in Fig. 7. While some of the OA composition variability could be visually extracted from Fig. 7, we focus on the description of Figs 8–10, which summarize the temporal behaviour of each OA component. The three components
explained ca. 70–80% of the OA variation at SMEAR II (Fig. 8a). The unexplained variation can be split into data with low SNR (noisy) and data with high SNR. The unexplained fraction due to high noise (low SNR) was lowest in summer, ca. 10%, otherwise ca. 20%. The rest of the unexplained OA variability (data with high SNR) was nearly constant at 10–12%. This fraction is termed as the "real unexplained variation" and includes only the variation made up by variables having the unexplained variation fraction less than 25% (Paatero, 2004).





### 6.2.1 LV-OOA

LV-OOA was always the dominating OA type at SMEAR II, both in terms of OA mass fraction ($f_{LV-OOA}$; Fig. 8b) and absolute concentration (Fig. 7a&9a). LV-OOA is understood to form as a result of OA aging in the atmosphere (e.g. Jimenez et al., 2009). Indeed, several OA types have been shown to chemically transform to LV-OOA in relatively short time scales (e.g. Jimenez et al., 2009). This makes the dominance of such aged OA product perfectly reasonable at a rural background site, such as SMEAR II. LV-OOA made up ca. 60% of OA mass concentration, and the median absolute LV-OOA loading was 0.74 µg m$^{-3}$ (overall LV-OOA IQR 0.35, 0.74, 1.46 µg m$^{-3}$).

LV-OOA loading had a bimodal seasonal cycle. The first peak occurred in February (February LV-OOA IQR: 0.30, 0.64, 1.28 µg m$^{-3}$), similarly as previously reported SMEAR II NR-PM$_1$ inorganics (Heikkinen et al., 2020). We previously speculated that this February peak of NR-PM$_1$ inorganics could result from a combination of meteorology-driven phenomena, such as more southerly winds compared to other winter months, the enhanced amount of solar radiation enabling photochemistry, or relatively dry conditions (in terms of less precipitation) diminishing wet deposition of aerosol particles upon transport from more polluted areas. Similar phenomena could certainly favour also higher LV-OOA loading in February. While LV-OOA mass spectrum does not offer insights of possible LV-OOA sources (spectrum comprises mostly of $m/Q$ 44 Th; Fig. 5a), we can still assume the wintertime LV-OOA sources to be mostly anthropogenic due to reduced biogenic activity in the wintertime boreal environment. Wintertime LV-OOA could be to a large extent for example aged wood-burning organic aerosol as wood burning is expected to be the most dominant wintertime OA source in Europe (Jiang et al., 2019). Also anthropogenic SOA formation in urban plumes is a potentially high source of wintertime OOA (Shah et al., 2019). Despite the less efficient oxidation (OH radical concentration much lower in wintertime compared to summer), the cold wintertime temperatures enable condensation of less oxidized organic vapours (e.g. (Stolzenburg et al., 2018)), which could favour wintertime SOA formation. Due to aging processes, it is likely that such wintertime (anthropogenic) SOA would be detected as LV-OOA at SMEAR II due to OOA aging during transport from the far-away urban plumes. The diurnal cycle of wintertime LV-OOA showed no diurnal pattern (Fig. 10a). Such behaviour is typical for long-range transported, i.e. not locally produced air pollutants, as boundary layer dynamics will not influence their concentration in the surface layer.

The second, yet most significant peak of LV-OOA loading occurred in summer (summertime LV-OOA IQR: 0.65, 1.18, 2.01 µg m$^{-3}$; Fig. 9a), when biogenic emissions rapidly produce SOA in ambient air. It is likely that in summertime biogenic processes were the dominating sources of LV-OOA. LV-OOA possessed a diurnal cycle clearly only in summer, where the LV-OOA reached a maximum concentration during daytime (Fig. 10a). It is likely that in contrary to wintertime, LV-OOA was produced also locally via photochemical pathways during daytime.



### 6.2.2 SV-OOA

The highest SV-OOA OA mass fraction ($f_{\text{SV-OOA}}$ Fig. 8b) and loading (Fig. 7b&9b) were observed in summer (unimodal seasonal cycle). The summertime $f_{\text{SV-OOA}}$ was ca. 40% (summertime SV-OOA IQR: 0.33, 0.59, 1.07 µg m$^{-3}$), otherwise ca. 25–30% (wintertime SV-OOA IQR: 0.10, 0.17, 0.28 µg m$^{-3}$). The seasonal cycle of SV-OOA could be explained by the
surrounding forest's enhanced biogenic activity in summer months, which leads to biogenic SOA formation. However, we are not able to confirm whether all of the SV-OOA is of biogenic origin. Importantly, because the nearby sawmills in Korkeakoski (ca. 7 km NE of SMEAR II; Sec. 2.1) represent significant SV-OOA plume sources (e.g. Äijälä et al., 2017). It is likely that SV-OOA production from terpenes emitted from the Korkeakoski sawmills also express seasonality following the air's oxidation capacity. In addition, it is also possible that terpene emissions from the Korkeakoski sawmills are also temperature-
dependent.

SV-OOA possessed a diurnal cycle in all months but December and January. The SV-OOA diurnal cycle was typical for semi-volatile species: the maximum loading was achieved in early mornings (Fig. 10b), when atmospheric mixing layer is typically the shallowest and temperature the lowest. We previously reported a similar seasonal cycle for NR-PM$_1$ nitrate at SMEAR II
(Heikkinen et al., 2020). The SV-OOA formation is likely strongly linked to the accumulation of monoterpenes in these shallow nocturnal boundary layers in forests. During calm, stable nights radiative cooling promotes formation of inversion layers hindering vertical dispersion of the forest's emissions. The cooling of the air enables partitioning of less-oxygenated gaseous species yielded from monoterpene oxidation to the condensed phase enhancing also SV-OOA formation. SV-OOA formation via condensation of highly oxidized organic molecules (HOM, which commonly originate from monoterpene oxidation;
(Bianchi et al., 2019)), has been previously suggested to occur at SMEAR II's nocturnal boundary layer(s) (Hao et al., 2018).

It is important to mention here that if these ACSM measurements were conducted in a higher altitude, perhaps even a few tens of metres above ground level, such strong diurnal cycle would likely not have been captured. In addition, upon the development of the turbulent daytime boundary layer the SV-OOA yielded during the night time is does likely not play any
major role in the SV-OOA loading in the daytime boundary layer. The BVOC oxidation in the boreal forest is more efficient during daytime compared to night time (e.g. (Peräkylä et al., 2014)), which would mean a higher production of condensable vapours potentially forming SV-OOA during daytime.

When summing up SV-OOA and LV-OOA, we can see that summertime OA was nearly exclusively OOA (which is typically
a good approximation of SOA), and even in wintertime OOA organic mass fraction was ca 80%. High OA mass fractions of OOA in PM$_1$ have been observed all over the Northern mid-latitudes (Zhang et al., 2007).



### 6.2.3 POA

The $f_{POA}$ seasonal cycle was opposite to that of SV-OOA, with highest $f_{POA}$ achieved in wintertime (13%; Fig. 8b). The summertime $f_{POA}$ was 3% and the overall median ca. 6%. Interestingly, when comparing the overall median to $f_{POA}$ estimated

previously at SMEAR II, we observe much lower fractions. For example, Äijälä et al. (2019) report a HOA OA mass fraction of 6% and BBOA OA mass fraction of 21%. The sum of them, which should somewhat represent POA, is 21 percentage points higher than the mean $f_{POA}$ reported here. As the Äijälä et al. (2019) study was conducted with an AMS the data set should certainly better capture short-term pollution plumes compared to the ACSM, which has significantly lower time resolution and higher noise level. Another important fact to consider is that the Äijälä et al. (2019) study period is situated between years

2008 and 2010. It is possible that POA emissions have reduced since then, or the emissions were for some reason higher than usual between 2008 and 2010. Hints of such long-term reduction or higher concentrations in 2008-2010 at SMEAR II can be observed in the equivalent black carbon (eBC) concentrations. The eBC concentration between years 2008 and 2011 was nearly twice as high as between years 2013–2018 (Luoma et al., 2020). This could certainly explain some of the discrepancy between these studies.


In addition to the $f_{POA}$, also the absolute POA concentration peaked in winter (Fig. 7c&9c). The seasonal cycle resembles that of $NO_x$ shown in our previous work (Heikkinen et al., 2020), which in turn follows the cycles of atmospheric boundary layer height and temperature. Several phenomena can explain a larger wintertime $f_{POA}$: wintertime POA dispersed in a shallower atmospheric mixing layer compared to summer, and sources of POA are possibly greater in winter due to enhanced need for

residential heating and less of POA evaporation due to cold temperatures. In addition, POA wintertime aging to LV-OOA is possibly hindered compared to summertime, due to less efficient photochemical oxidation. The wintertime POA diurnal cycle showed most of the time a minor afternoon maximum and a minor night-time elevation was slightly visible only in late January/ early February (Fig. 10c). Typical HOA diurnal cycles in populated areas show an extremely distinct diurnal pattern following morning and evening rush hours (e.g. (Zhang et al., 2005)). In residential areas, BBOA in turn typically clearly peaks in the

evening, when domestic heating takes place and the emissions are dispersed in the nocturnal boundary layer (e.g. (Canonaco et al., 2013)). Due to SMEAR II's distance from major HOA and BBOA sources, we did not observe such clear POA diurnal cycles in neither summer nor winter. The summertime POA diurnal cycle resembled a diurnal cycle of the sum of LV-OOA and SV-OOA. As discussed earlier in Sec. 5.3.1, it is likely that summertime POA loading was overestimated by the rCMB-model (Fig. S.3).

### 6.2.4 Long-term trends

Due to the relatively long time series and decreasing trends in anthropogenic air pollutant concentrations in many populated areas around the world (e.g. (Zhang et al., 2019;Luoma et al., 2020;Wang et al., 2012;Zheng et al., 2018)), we wanted to explore whether any effects of such would be visible also at SMEAR II. To do so, we first applied a commonly used Mann-





Kendall test (MK-test) (Gilbert, 1987) to determine whether any significant trends existed (in summer and winter data,
respectively; Fig. S.6), and then calculated the possible trends using Sen's slopes (Gilbert, 1987).

Only winters in 2013 and 2018 contained two months of wintertime data (ca. 67% data coverage of the season; season = DJF),
whereas the rest of the winters only contained 1–1.5 months of measurement data per winter season. Such measurement gaps
give more room for biased trend estimations. A recent paper from (Zhai et al., 2019) highlights the importance of
meteorological anomalies in trend estimations in China when analysing a data set of similar length to ours. The authors related
12% of the 30–50% reduction in $PM_{2.5}$ observed across China in 2013–2018 to meteorology. As circulation patterns influence
weather at SMEAR II along the origin of the air mass to be detected, these patterns will also influence aerosol chemical
composition, which is tightly linked to meteorology at the site (Heikkinen et al., 2020).

The relationship between temperature and OA loading serves as a good example of the effect of meteorology on NR-PM$_1$
composition: the greatest OA loadings and mass fractions were recorded at SMEAR II during the warmest summers of the
measurement period (summers of 2014 and 2018; Heikkinen et al., 2020). Thus, during summertime when data coverage was
excellent (only years 2012 and 2016 were missing more than one month of summer data), the trend analysis is challenged by
the occurrence of warm summers. Eight years of summer measurement data is certainly not enough to achieve any good
estimation of the heat wave frequency in Southern Finland to analyse reliably summertime OA trends. Thus, a comprehensive
and reliable trend analysis at SMEAR II should involve an even longer data set, the analysis of which should be conducted
tightly together with meteorological analyses. Therefore, instead of analysing any trend magnitudes we report a minor
possibility for a decreasing summertime and wintertime LV-OOA and A-OA trends, and a minor possibility for an increasing
SV-OOA trend, but the reasons behind these would require more analysis. Most weight for these trends were given by years
2012 and 2013, when the ACSM was operated without a dryer, which could also play a role.






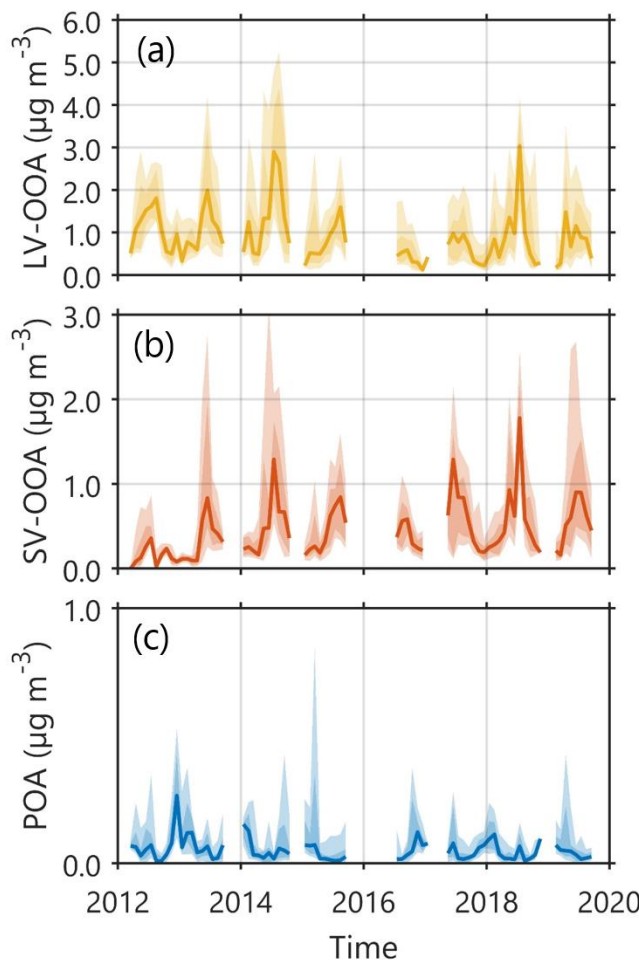

**Figure 7** Monthly resolution time series of LV-OOA (panel a), SV-OOA (panel b) and POA (panel c) mass concentrations obtained with rCMB. The light shadings indicate the area between the 10[th] and 90[th] percentiles, and the dark shadings the area between the 25[th] and 75[th] percentiles. The solid line represents the monthly medians for each month of measurements in 2012 – 2019. Note the different *y*-axes scales (grid lines are drawn every 1 µg m$^{-3}$).






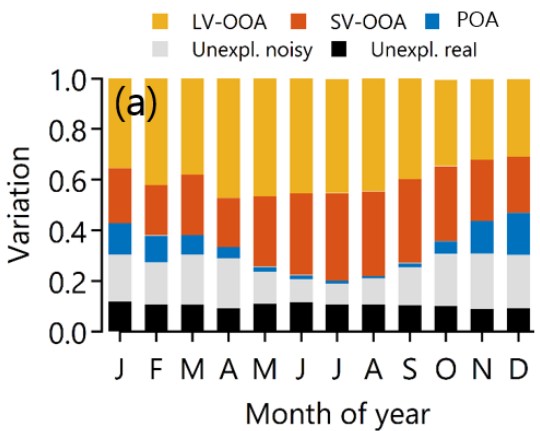
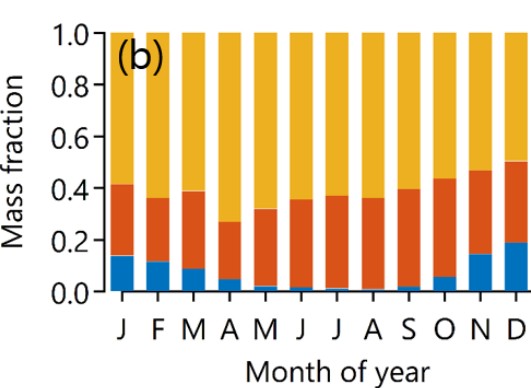

**Figure 8** The panel (a) depicts the variability of the rCMB compared to measurement variability (scaled by uncertainty). The unexplained
fraction is ca. 30% outside summer, when its ca. 25%. This variation in the unexplained variation is due to increased noisy fraction (light
grey) outside summer. The real unexplained fraction (in black) stays at rather constant of ca. 11%. Panel (b) shows $f_{LV-OOA}$, $f_{SV-OOA}$ and $f_{POA}$
in different months. This panel only visualizes their variability in rCMB.



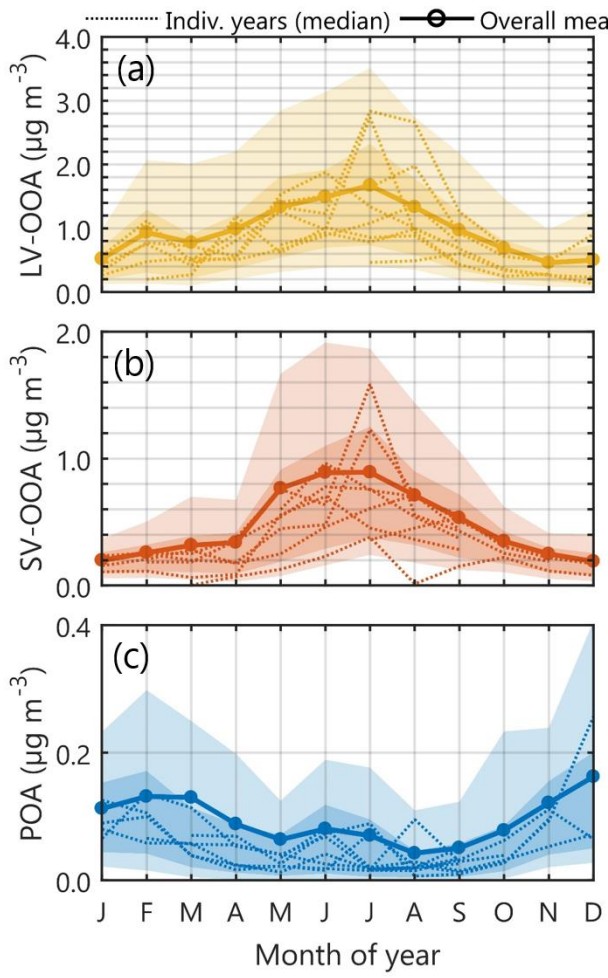

**Figure 9** The monthly mass concentrations of LV-OOA (panel a), SV-OOA (panel b) and POA (panel c) obtained with rCMB. The light shadings indicate the area between the 10th and 90th percentiles, and the dark shadings the area between the 25th and 75th percentiles. The narrow dotted lines represent monthly medians for individual years and the dark lines with circled markers represent the overall monthly mean concentrations. Note the different *y*-axes scales (grid lines are drawn every 0.2 µg m-3).



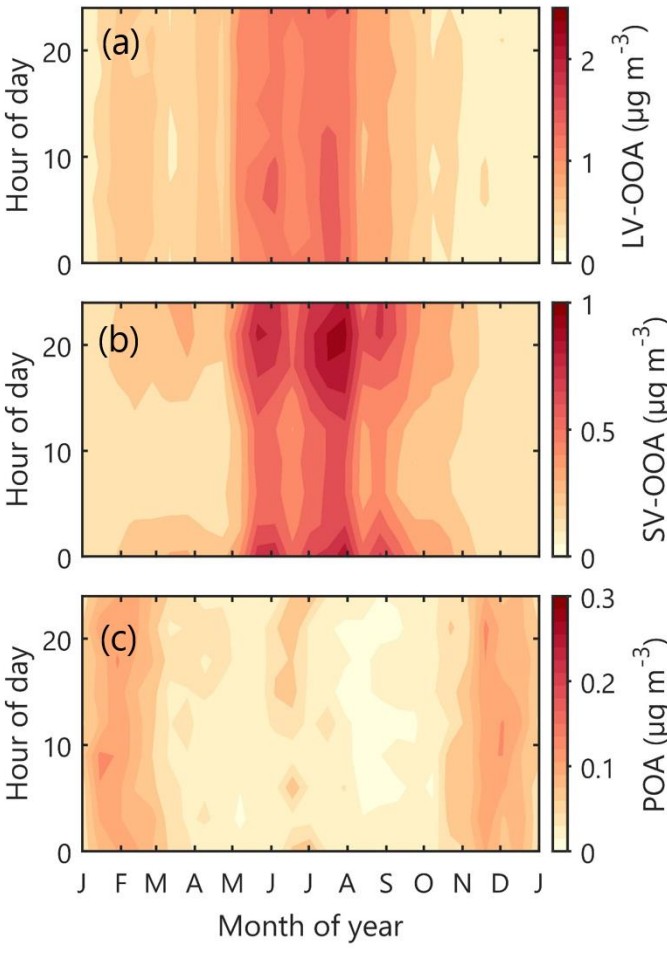

**Figure 10** The median diurnal cycles of LV-OOA (panel a), SV-OOA (panel b) and POA (panel c) obtained via rCMB. The y-axes represent
the local time of day (UTC+2) and x-axes the month. The colour scales represent the mass concentration of each OA type. Note the different
scales for each plot. Each grid point represents a 14d × 3h period, visualized with the MATLAB 2017a *contourf* function.



### 6.3 Wind and air mass trajectory influence on OA composition

In this section we will discuss the wind direction and speed dependencies of OA composition, which provide useful insights in estimating whether OA is locally produced or transported. After this analysis we briefly examine the OA types' behaviour as a function of time over land (Sec. 3) to understand the potential magnitude of natural aerosol formation over the boreal forest.

### 6.3.1 Openair polar plots

The Openair polar plot for LV-OOA is displayed in Fig. 11a. Based on this figure, elevated LV-OOA concentrations could be expected from SE (polluted sectors) regardless of the wind speed. In case of easterly winds, the LV-OOA concentrations were generally the highest if wind speeds stayed below 20 km h$^{-1}$ (ca. 5.6 m s$^{-1}$). On the contrary, in the case of NW winds (winds from the clean sector) with wind speeds exceeding 20 km h$^{-1}$, the LV-OOA concentration approached zero implying clean air transport. The LV-OOA Openair polar plot resembles greatly the overall NR-PM$_1$ organics' Openair polar plot visualized
previously in Heikkinen et al. (2020), which was also expected due to LV-OOA being the dominant OA component. The LV-OOA Openair polar plot had more southerly influence in wintertime (Fig. 11b) and significantly less LV-OOA was detected with SE winds compared to the overall picture (Fig. 11a). The summertime LV-OOA Openair polar plot (Fig. 11c) in turn was nearly identical to the median plot including all months.

The SV-OOA concentration was highest with low wind speeds (below 10 km h$^{-1}$, i.e. ca. 2.8 m s$^{-1}$; Fig. 11d). In addition, SE winds favoured SV-OOA presence. As SV-OOA loading peaked at night (Fig. 9c), the low wind speed dependence of SV-OOA indicates that calm nights are most suitable for SV-OOA detection. Low nocturnal wind speeds promote the formation of shallow nocturnal boundary layers, as the mixing is not enhanced by mechanically produced eddies. Thus, both the SV-OOA diurnal cycle and the SV-OOA formation boost at low wind speeds support the hypothesis that SV-OOA is produced
locally and it builds up in the night time surface air. However, the Korkeakoski sawmills probably explain why SV-OOA concentration field is darker at the SE side of the Openair plot origin (Fig. 11d). The wintertime SV-OOA Openair polar plot still showed highest SV-OOA loading with low wind speeds, however having less SE influence in the concentration field (Fig. 11e). The summertime polar plot (Fig. 11f) again resembled the overall plot (Fig. 11f). This summertime concentration field of SV-OOA greatly resembled the summertime LV-OOA concentration field (Fig. 11c). The Pearson correlation coefficient
between these fields was $R = 0.87$. This similarity supports the previously stated hypothesis that summertime LV-OOA was likely of biogenic origin (also with possible sawmill influence).

Finally, the POA Openair polar plot (Fig. 11g) exemplifies how specific wind direction and speed combinations were required for POA detection: POA was resolvable only if the wind direction was S –SE and wind speed ca. 20 km h$^{-1}$ (rarely the case at





SMEAR II; Fig. S.1). While such high wind speeds ultimately reduce the time the air masses spend over populated areas with potentially high POA emissions, the high wind speeds also enable fast transport of the POA types making their detection at fresh state possible (before POA has aged or evaporated).

The wintertime POA Openair polar plot had also SE influence with less high wind speeds (Fig. 11h). It greatly resembled the
wintertime LV-OOA Openair polar plot (Fig. 11b). The Pearson correlation coefficient between the wintertime POA concentration field and LV-OOA concentration field was $R = 0.93$. The high agreement between these concentration fields supports the previously stated hypothesis that wintertime LV-OOA was likely of anthropogenic origin. The summertime POA Openair polar plot (Fig. 11i) was not greatly differing from the other POA Openair polar plots, which gives some confidence in summertime POA quantification: if most summertime POA was overestimated, the summertime POA Openair polar plot
would likely have similar wind dependence as OOA.

### 6.3.2 Time over land analysis

Tunved et al. (2006) showed how (organic) aerosol mass concentration increased as a function of time over land (TOL; i.e. the number of hours the air mass spent over the forested land surface upwind of SMEAR II) when the land surface had little anthropogenic influence (e.g. in the clean north-westerly sector; Fig. S.2). This increase was attributed to natural (biogenic)
OA production in the boreal boundary layer. Here, we observe a similar increase in the clean sector (Fig. 12a), LV-OOA loading being the most sensitive to TOL (Fig. 12). The lower increase shown for SV-OOA (Fig. 12) in comparison to LV-OOA supports our hypothesis of SV-OOA sources being also local and SV-OOA aging into LV-OOA. The relationship between POA and TOL was not significant (Fig. 12). The increase of LV-OOA loading as a function of TOL indicates OA formation in the boreal boundary layer, its build-up in the air mass, and aging into LV-OOA prior to arrival at SMEAR II.
Such phenomenon is not visible when investigating the OA types' behaviour as a function of TOL in polluted sectors (Fig. S.7). Indeed, none of the OA-types indicate links between OA loading and TOL in neither air masses of European (southerly sector) nor Russian (easterly sector) origin. We are not surprised of such lack of correlation between OA and TOL as the picture is greatly hampered by anthropogenic emissions. As the anthropogenic emissions are minor in the clean sector, and as suggested by Tunved et al. (2006), the OA production in the clean sector is dominated by biogenic SOA formation.


The biogenic SOA hypothesis is supported also by the seasonality of the OA vs TOL relationship (Fig. 12b): a highest correlation between the two and the steepest OA increase as a function of TOL is observed in July, which held the greatest temperatures during the measurement period (Heikkinen et al., 2020). Such temperature dependence is typically associated with biogenic SOA production (e.g. (Daellenbach et al., 2017;Stefenelli et al., 2019)) as the emission rates of several SOA
precursors (such as monoterpenes) increase as a function of temperature (Guenther et al., 1993). The linear regression slopes for a LV-OOA vs TOL scatter plot would suggest LV-OOA formation of ca. 42 ng m$^{-3}$ h$^{-1}$ in July, which is twice the SV-OOA vs TOL slope (Fig. 12b). To exemplify these numbers, three days over the boreal forest in July would yield ca. 3 µg m$^{-3}$ of





LV-OOA and 1.6 µg m$^{-3}$ of SV-OOA. The slopes for LV-OOA stay below 10 ng m$^{-3}$ h$^{-1}$ between October and April (values
similar to the slopes for SV-OOA at the same time; Fig. 12b), when there is less of biogenic plant activity. These slopes were

similar in magnitude to those derived previously for SMEAR II data (Tunved et al., 2006;Liao et al., 2014). Another interesting
feature extracted from this analysis was that if the OA type vs TOL slopes were calculated using data only below TOL = 40 h,
the SV-OOA and LV-OOA slopes would be identical, and only after TOL exceeds 40 h, LV-OOA loading keeps increasing
while the SV-OOA loading shows a minor decreasing trend (Fig. 12c). More analysis and perhaps investigations of similar
plots from other boreal research stations could give us insights whether the figure informs more of time scales of OA chemistry

or whether it is linked to meteorology and/or distance to the ocean from the measurement station.

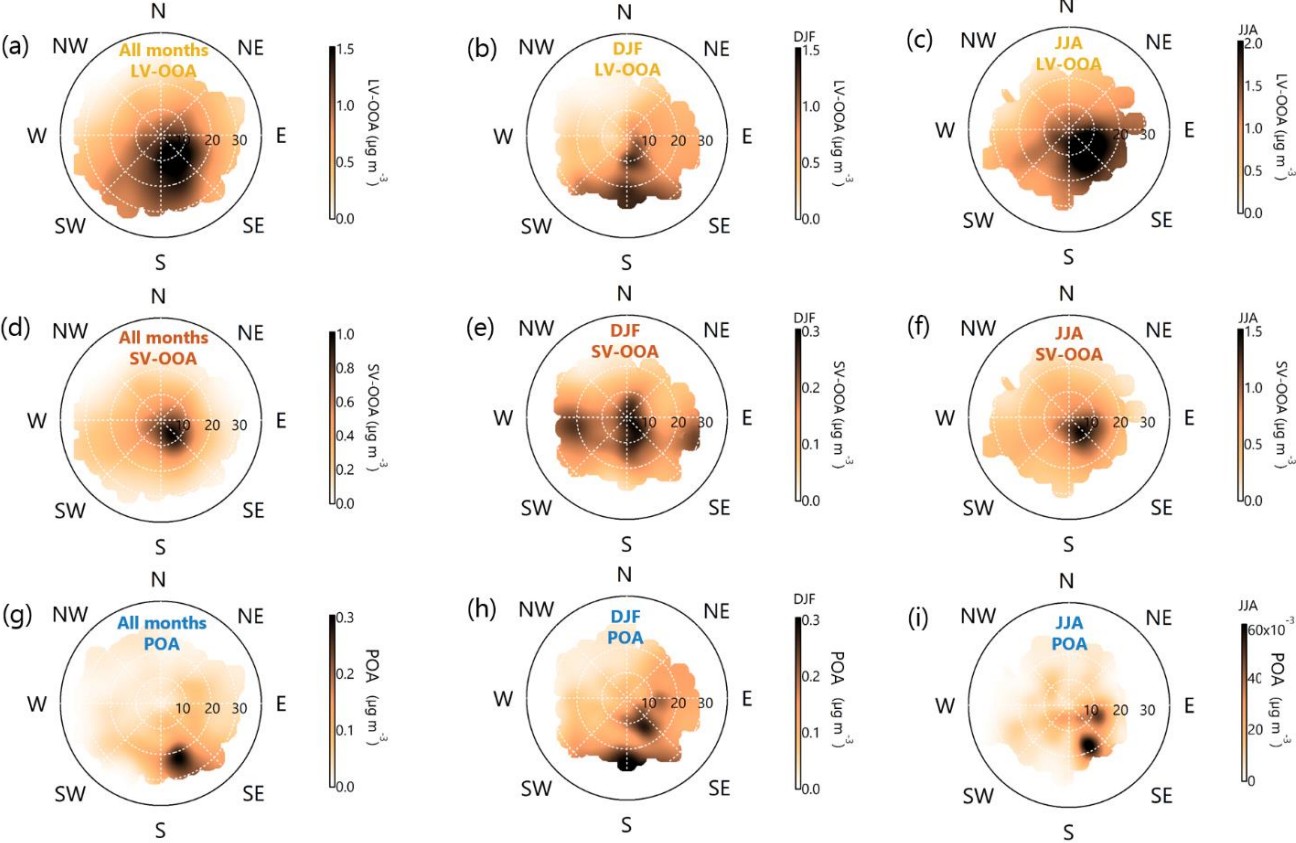

**Figure 11** Openair polar plots (Carslaw and Ropkins, 2012) for LV-OOA (first row), SV-OOA (second row) and POA (third row) obtained
via ZeFir pollution tracker Wavemetrics Igor Pro toolkit (Petit et al., 2017). The first column represents the median over all seasons, the
second column the median over wintertime and third the median summertime. The distances from the circle origins indicate wind speeds (in
km h$^{-1}$). Wind speed grid lines are presented with dark grey dashed lines. The colour scales represent the mass concentration of each OA
type modelled via rCMB during the specific wind direction and speed combinations. Note that the scales are different among the subplots.
As these figures, do not indicate any likelihood of these wind direction and speed combinations, Fig. S.1 is important to keep in mind while
interpreting them. Briefly, N-NE-E is the least likely direction of wind, and S-SW-W is the most likely. Wind speeds rarely exhibit 20 km
h$^{-1}$. The wind direction and speed data are collected above the boreal forest canopy.





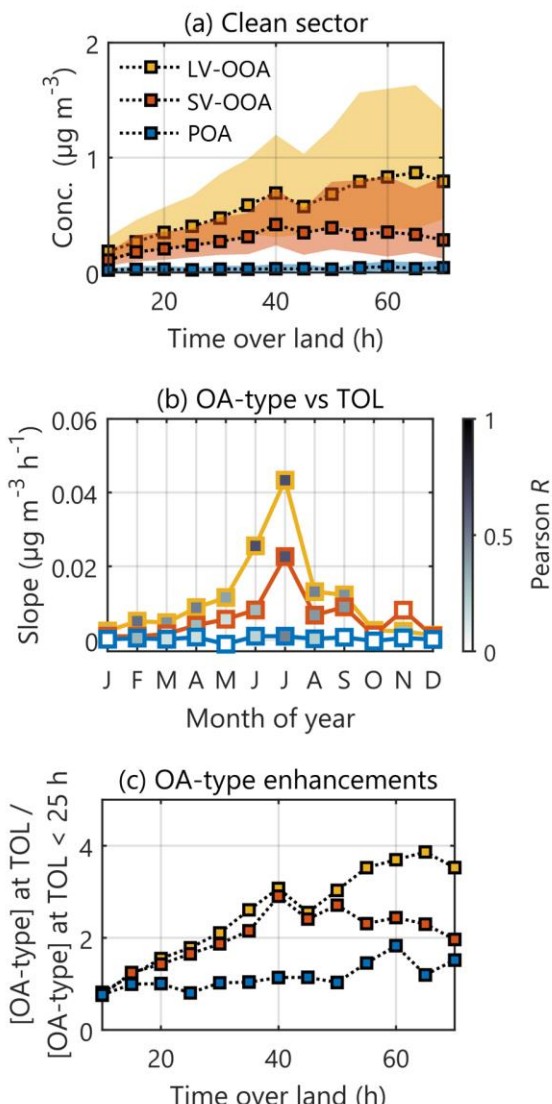

**Figure 12** (a) The different rCMB factors ($y$ axes in µg m$^{-3}$) vs TOL ($x$ axes in hours) for the clean sector (least polluted north-western sector as defined by Tunved et al., 2006; see Fig. S.2 for a more precise sector definition). The data are binned to 5-hourly TOL bins. The shaded areas represents the concentration interquartile ranges (25$^{th}$ to 75$^{th}$ percentile) and the square markers the median concentrations. (b) The
slopes (in µg m$^{-3}$ h$^{-1}$) are calculated for a linear fit between TOL ([20, 70] h) and the three different OA types. (c) The OA type concentration in the TOL bin divided by the median OA type concentration when TOL was < 25 h as a function of TOL. The plot visualizes how the SV-OOA and LV-OOA have similar behaviour until TOL = 40 h.



## 7 Conclusions

Organic aerosol (OA) mass spectra are recorded continuously with an Aerosol Chemical Speciation Monitor (ACSM) since 2012 at SMEAR II station, located within the boreal forest in Southern Finland. The goal of the current paper was to yield understanding of the main OA components: their mass spectral features and temporal behaviours. The large extent of input data (eight years) required us to develop a new framework for conducting OA chemical characterization, as to our knowledge there are no previous studies where equally long or longer time series of OA mass spectra have been characterized. We approached the OA characterization via Positive Matrix Factorization (PMF; Paatero and Tapper, 1994). However, due to the length of the data set, we conducted the PMF with a 30-day rolling window approach, which enabled factor profile variability across the eight years (Canonaco et al., 2020;Parworth et al., 2015). The rolling PMF yielded an extremely large number of PMF solutions (20 900 solutions, 62 700 factor profiles), and the manual exploration of these would not be feasible. Instead, we utilized a machine learning approach where these rolling PMF solutions were managed via K-Means clustering, as an alternative data sorting technique to the more commonly used criteria-based selection of solutions (Daellenbach et al., 2017;Vlachou et al., 2019;Canonaco et al., 2020) , which is typically based on time consuming correlation analyses. With our approach, we extracted three significantly different OA clusters: low-volatility oxygenated OA (LV-OOA), semi-volatile oxygenated OA (SV-OOA) and primary OA (POA). By anchoring a CMB-type of PMF run with these OA cluster centroids and their intra-cluster variability, we were able to explain ca. 70% of the observed OA at SMEAR II. Importantly, nearly two thirds of the unexplained variation was due to high noise level of the data leaving the real unexplained variation at only 11%. The analysis method utilized here turned out to be robust, and it required little analyst interference. Therefore, our framework presents a technique to effectively analyse long-term AMS or ACSM datasets while reducing subjective bias upon analysis. As several similar continuous aerosol chemical composition measurements already take place in multiple locations in the world, we expect that the need of this type of machine learning driven analysis procedures to increase in the future with the increasing amounts of data.

With equal importance to the developed data analysis framework, we also presented the OA composition and its variability at SMEAR II. The main conclusion to be drawn from the OA composition at SMEAR II is that this boreal OA is nearly exclusively oxidized organic aerosol, mostly highly oxidized LV-OOA. The result was well in line with previous studies from the Northern Hemisphere showing the ubiquity of OOA especially at rural measurement sites (Zhang et al., 2007). The LV-OOA seasonal cycle was bimodal culminating in February and summer. The wintertime LV-OOA was likely anthropogenic and the February peak coincided with NR-PM$_1$ inorganics (Heikkinen et al., 2020). The summertime LV-OOA had enhanced biogenic influence and it was linearly increasing the longer the air mass had spent over the boreal forest. We estimated natural LV-OOA production of several tens of ng m$^{-3}$ per hour. These numbers were well in line with previous studies investigating

the natural aerosol production in the boreal forest (Tunved et al., 2006;Liao et al., 2014). SV-OOA was the second most abundant OA type and the maximum SV-OOA concentration was detected in early mornings during summer. Both biogenic processes and emissions from the nearby sawmill contribute to the SV-OOA mass as also exemplified in previous studies (e.g. Äijälä et al., 2017). Highest SV-OOA loadings were observed when sampling from shallow nocturnal surface layers, but it is

possible that the production of SV-OOA was highest during daytime when most BVOC oxidation takes place. Finally, the POA, the mass spectrum of which resembled both hydrocarbon-like OA and biomass burning OA, attained significant OA mass fractions only in winter. Still, those OA mass fractions were significantly lower compared to earlier long-term descriptions of SMEAR II OA composition (Äijälä et al., 2019). This discrepancy could be for example linked to a decrease in POA emissions as hinted by decreasing BC trends at the site (Luoma et al., 2020), or the ACSM limited capability in

detecting short-term (pollution) plumes, which average out even more due to the 3-hour averaging applied to the PMF input data, which was necessary to improve the SNR at this rural background site. More generally, due to OA composition sensitivity to meteorological conditions and anomalies, even longer time series need to be accumulated in order to reliably estimate trends of POA and other OA constituents at SMEAR II based on ACSM data.

**Data availability**

The ACSM NR-PM$_1$ OA concentration data are available at EBAS database under EMEP ACTRIS framework as well as upon request from the corresponding authors. The PMF matrices and OA classes' mass spectral profiles and time series are available upon request from the corresponding authors. The wind direction and speed data are available at the SmartSMEAR data repository (https://avaa.tdata.fi/web/smart) (Junninen et al., 2009). Contact of the original data contributors can be requested from atm-data@helsinki.fi.

**Competing interests**

The authors declare no conflict of interest.

**Author contributions**

LH, MÄ, ME, TP, MK, and DW designed the study. LH, MÄ and FG performed the ACSM measurements. PA provided size distribution data needed for 2019 ACSM data processing. MR performed time over land calculations with HYSPLIT

trajectories. KL provided BC data for 2019 ACSM data processing and assisted LH with other trend calculations. The ACSM data processing was performed by LH. KD, CG and MÄ assisted LH with (rolling) PMF. MÄ and DA assisted LH with the K-Means clustering. LH performed the overall analysis, data visualisation and wrote the paper with comments from the co-authors. ME supervised all the steps in this process.



**Acknowledgements**

We thank SMEAR II staff  for keeping the measurements running, COST COLOSSAL for valuable guidance and discussions,
Francesco Canonaco (Datalystica Ltd) for PMF support. We thank Santtu Mikkonen, Jean-Eudes Petit for useful discussions.
The European Research Council Horizon 2020 (grants 638703, 689443, 821205), the Academy of Finland (grants 317380,
320094, 307537, 324259, 333397 and 334792) supported this research. Finally, we acknowledge the University of Helsinki
and Academy of Finland support to ACTRIS infrastructure (grants 329274 and 328616).

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
