# Peer review of "Eight years of organic aerosol composition data from the boreal forest characterized using a machine-learning approach"

_Atmospheric Chemistry and Physics, 2020_

## Referee Comment (RC1) · Anonymous Referee #1 · 24 Sep 2020

This paper presents a factorisation analysis of a very long time series of ACSM data from the Hyytialla site. While PMF analysis has worked well for short AMS and ACSM datasets, multi-season and multi-annual datasets have previously necessitated breaking the dataset down into small chunks or using a 'rolling window' method. The approach here is to use a 'rolling relaxed CMB' approach, with the a priori mass spectral profiles generated using the unsupervised clustering of bootstrapped 'rolling PMF'. This is relevant for ACP, because while this paper does have a heavily technical bent, it also reports general interpretations of the driving factors behind aerosol behaviour at this

heavily-studied site that may have implications for other studies.

While this paper presents a very detailed account of what was done in terms of analysis, I do not find myself completely convinced that the authors have sufficiently argued the case for why this approach should be considered superior to other existing techniques. To be clear, I am not necessarily saying that they should have done anything differently (I would particularly commend the adherence to unsupervised methods for the sake of objectivity), but my opinion is that the authors need present stronger arguments for why certain choices were made in the approach to data analysis and they also need to go further in exploring the strengths and weaknesses of this approach (compared to others) rather than simply accept the outputs at face value. It's also not completely clear to me how different the interpretation of the results has been positively influenced by the use of the new technique. Put simply, it's not very clearly spelled out what the authors were trying to achieve with this approach, what the underlying principles were (as opposed to the detail of the algorithms), and whether they objectively succeeded in meeting their original objectives. For these reasons, I am recommending publication subject to major revisions. I would ask the authors to consider the points below.

Major comments:

It took me several reads before I really thought I understood the philosophy behind this technique, as there are a frankly baffling number of analysis stages. The authors need to be much clearer in how they describe the approach and I don't think presenting it in a stepwise form in figure 1 really helps. Instead, it should be made clear from the beginning that (assuming I did understand it correctly) the ultimate goal is to generate factorisation outputs using rolling rCMB and the biggest emphasis should be placed on this technique. As far as I can tell, all of the other stages leading up to this are merely generating objective mass spectral profiles that this algorithm uses as an input.

The use of clustering represents the part of the analysis that is traditionally used least

in ACSM analysis and therefore represents the part with the least precedent. Certain decisions are made at various stages of the clustering and while the authors present arguments for why these would be considered reasonable, they do not really explore the notion of what would happen if they had done it in a different manner. I would specifically point to: 1. The use of k-means as a clustering algorithm. 2. The use of cosine angle rather than Euclidian distance as the distance measure. 3. The weighting of higher mass to charge ratios. 4. The choice of methods used to determine the optimum number of clusters and centroid profiles. While I am not questioning the individual choices, I would expect the authors to offer more reasons why alternatives were discounted, reporting on any undesirable behaviour when alternatives were test, where available.

I would consider the benchmark comparison of this technique to be rolling PMF, as this is an already-developed technique that is used to interpret long ACSM datasets. In figure 4, the authors compare weighted residuals and the reader could be forgiven for thinking that this has resulted in an inferior data product to what was obtained at the end of stage 1 of the analysis. The authors need to discuss in more depth the pros and cons of each approach and present a stronger case for why it should be considered advantageous to use the technique used here. This should bear in mind that just because a particular algorithm uses less supervision and produces a less ambiguous result, these do not in themselves mean the data products are intrinsically more accurate.

As part of any numerical data reduction such as this, it is vital to properly explore its limitations and I don't consider what is presented here to be sufficient. I am particularly interested in the analysis of residuals, which is normally the first thing to inspect. Is the unexplained variance and mass shown in figure 8 purely random noise, or is there any structure (relative to m/z, season, temperature, time of day, etc.) that might suggest there are factors at work that this does not adequately capture?

A key detail in the final data products is by how much the mass spectral profiles of the

factors was allowed to vary in the rCMB analysis. As I understand it, if the profile could vary with the rolling window, then this would avoid many of the limitations imposed by the default PMF data model, however this then creates implications for how the results are interpreted, in particular with the seasonal analysis and long term trends. Can the authors be sure that any interpretations presented in section 6.2 are the result in changes in the abundance of the different organic aerosol types, or changes in the mass spectral profile? If, on the other hand, the profiles are rigid, then this instantly asks questions of whether the factorisation is equally applicable at all times, or whether the technique is still susceptible to the same rotational ambiguity problems as conventional PMF.

Generally speaking, the explanations of the behaviour in section 6.2 are largely speculative and seem to be focused on creating a plausible narrative rather than offering new scientific insight. The authors should focus the discussion on what this new work adds to the (already substantial) body of work concerning this site and tropospheric aerosol processes in general, specifically through the virtue of adopting this new technique (as opposed to other existing methods). While hypotheses are frequently referred to casually in the text, it's not completely clear how these are being tested by the data and to what certainty.

Minor comments:

The term 'openair' is used to identify certain types of plots, but openair represents a large suite of many different graphing tools. Furthermore, openair wasn't actually used to generate the ones here. While it may be appropriate to credit the development of openair with popularising these graph types, the plots should be referred to by their specific type, e.g. polar plots.

I found section 6 overly wordy, with more text than was necessary to convey the important information. The authors may wish to cut back on the amount of discussion presented concerning the approach to analysis, instead focusing on conveying the

new scientific insights this offers, ideally in the form of hypothesis testing.

One of the problems that has traditionally confounded long term PMF analysis is that the mass spectral profiles of OOA factors can vary with season. Did the authors find any evidence of this at any stage of the analysis?

Regarding figure 6, the overlaid ellipses are not necessary, given that the points are already coloured. I would remove them, as they only serve to distract.

Also regarding figure 6, a number of points associated with LV-OOA have a high f60, which is a classic symptom of LV-OOA 'mixing' with BBOA, owing to the high HULIS content of the latter. The authors should comment on this.

Section 6.2.4: A statistical treatment is referred to, but no actual quantitative results are presented. This should be done, even if it is to report that no significant trend was observed.

---

## Referee Comment (RC2) · Anonymous Referee #2 · 18 Jan 2021

Atmospheric Chemistry and Physics Manuscript ID: acp-2020-868 Title: Eight years of sub-micrometre organic aerosol composition data from the boreal forest characterized using a machine-learning approach

General comments:

This manuscript describes an interesting approach to deal with the source apportionment of long submicron organic aerosol (OA) datasets at a remote site (SMEAR II research supersite in the boreal forest). At these types of sites, far from primary sources,

a "classical" rolling PMF method (commonly based on the identification and constraining of primary factors through a criteria-based approach) may not be sufficient to deal with mostly only secondary factors.

This article presents a new methodological approach based on K-means clustering of a very large number of unconstrained rolling PMF solutions, to overcome this difficulty. The extracted clusters are further used to run a CMB-type (constrained) rolling PMF. Output results are finally utilized to provide an overview of the temporal variability of the three OA factors identified. Methodological results (section 5) extend from pages 11 to 21 (with about 3 pages of Figures), while the "Results" section (section 6) ranges from page 21 to 37 (with about 7 pages of Figures). The general impression is that the paper permanently hesitates (and authors have been unable to choose) between the two focal points (methodology- or result-oriented) which is confusing and frustrating at times, especially when the methodological choices seem too descriptive (see specific comments below).

The overall quality of the manuscript is excellent. A few minor technical corrections are reported at the end of this document.

\*\*\*\*\*

Specific comments:

L40-41: "it is also possible that ACSM was less efficiently capturing short term (POA) pollution plumes". I believe this assumption derives from the behavior observed with the 3-hour averaged dataset. Therefore could it rather arise from the averaging (diluting the information over a longer time window) compared to the ACSM raw time resolution (and even more if compared to high time resolution AMS measurements performed at the same site) rather than from sampling differences like "capturing" may suggest?

L171 & L174: "Open air (...) analyses" and "Openair polar plots" are not informative titles and should be modified. Maybe something like "Influence of meteorological pa-

**ACPD**
rameters on air pollutant concentrations" for section 3, and "OA variation with wind direction and speed" for subsection 3.1 (or something along that line)? See also section 6.3.1 (L789).

L183: In the backtrajectory and TOL analysis, were the possible rainfalls along the air mass trajectory taken into account since it could highly influence the aerosol loading arriving at the site?

L323: Considering this is a new method, this section should not only be descriptive, but also provide some elements to compare between unconstrained rolling PMF/constrained rolling PMF/machine-learning approach ... and highlight the advantages of this latter methodology – at least in this case – to provide a more accurate analysis of the OA fraction. It is not so obvious to me after reading the manuscript what logic was behind the different validation steps (which are not so explicit).

L330: Since a correct assessment of the error matrix is critical for PMF analysis, please describe (at least in the SI) how the error propagation was performed for the 3-hour averaged OA.

L332: The most common approach uses a step function to downweight the weak and bad variables, based on averaged values for each m/Q. Although I agree with the authors that the cell-wise function may be more appropriate, especially if there is some strong seasonal variability for specific m/Q, have you tested both types of weighing to estimate the difference?

L339: If I understand correctly, since the initial rolling PMF is unconstrained, it is possible to have three factors that vary over the seasons/years. For instance, BBOA, HOA and 1 OOA in winter then switching gradually to HOA and 2 OOAs. But what if only two factors are relevant for some runs? Did you have a way to account for that (and eventually discard those runs)? It is relatively common in PMF analyses to keep only the "best" runs and discard those who do not fit criteria. Here I am not sure to understand if that happens in that first step (I believe not) or if using the weighted cluster centroids

**ACPD**
in the second phase is indeed taking care of the possible outliers.

L359 (Figure 1): I liked the summary of all the steps in that Figure but it would be interesting to make the link with the corresponding sections in the text, either directly on the Figure or at least in the caption.

L394: There could also be strong arguments against mass-to-charge scaling for this type of measurements. Because of their low signal-to-noise ratios, high m/Qs are often downweighted (as weak variables) in the PMF analysis, even if they can contain more information on OA sources. Have you selected only those for which the SNR was high enough in the dataset? Otherwise it seems contradictory to the previous reasoning applied to PMF. And I am wondering if it could play a role in the slight overestimation of POA in summertime when applying the rCMB.

L521 and following, and Fig. S.4: What is the proportion of "the PMF windows where POA was not classified"? The number of runs for which this factor appears compared to the two others could be indicated in Fig. S.4 as well (I guess it is N = 28, mentioned in Fig. 5e which appears later in the text?). What is puzzling me is that POA should appear more "easily" in the rCMB PMF runs when its concentration is higher, and clearly this is what will drive the slope of the linear fit in Fig. S.4c. Here the slope between the two POA factors (from clustering and rCMB) stays low (0.48) despite quite a few points at high concentrations.

L650: Stronger conclusions on the sources (anthropogenic or not; local or not) of LV-OOA could be drawn from various wind data analyses (backtrajectories, NWR plots, etc.). Thus it feels strange to find them here since section 6.3 specifically deals with the wind sectors influencing the sampling site. I would suggest to shorten the discussion here and refer to section 6.3 where you should be able to give stronger conclusions.

L713: "a larger wintertime fPOA:". Confusing. I thought the discussion had switched to POA absolute concentrations and not fPOA anymore?
L725: I am not convinced by the usefulness of this section, which I feel unconnected to the rest of the story. The authors themselves state that the dataset is probably not long enough to conduct a robust trend analysis, and that their assessment is likely biased by missing data for some seasons, or technical changes in the sampling line (no dryer for the two earlier years). Besides, this is clearly indicated in the conclusion as well (L916-918).

**\*\*\*\*\**

Technical corrections:

- L33: "However, also the nearby". Please consider revising this sentence.
- L114: "recorded"
- L119: "a new framework"
- L166: Please provide the CPC model.
- L238-240: Unclear. Please consider revising this sentence.
- L297: "Reassigning all the points"
- L317: "indicate a good"
- L322-323: "a relaxed (...) PMF analysis" (or another singular term)
- L347: "28 days". I believe it is not hours.
- L347: "Only window widths"
- L367 (Figure 2): please specify "time series of 3-hour averaged OA"
- L509: "such a dynamic"
- L512: remove "rolling"
- L591-592: "we can see that the our SV-OOA O:C". Delete "the".
L627-628: "in in". Delete one.

L663 and other occurrences in the text: "diel cycle" would be a more appropriate term than "diurnal cycle".

L671-672: seems like a sentence fragment. Consider revising.

L689: "is does likely not play". Please revise.

L748: "A-OA". Do you mean "POA"?

L757 (Figure 7): I honestly do not see a great difference between the light and dark shadings. Could you add more contrast?

L764: "POA" in the caption does not seem vertically aligned with the others.

L863: "As these figures, do not indicate". Delete the comma.

L869 (Figure 12a): IQR overlap in this plot. May be more readable with one plot per factor.

L895: "that the need (...) to increase". Please revise.

L1225-1230: Two references not in alphabetical order.

Supplementary information

L33: "The coloured lines represent"

Figure S6: Plots would be more readable if vertically expanded. Coloured and grey shaded areas are not explicit in the caption.

Figure S7: IQR overlap in these plots. They may be more readable with one plot per factor.

**ACPD**

---

## Author Response (AR1)

*Dear reviewers,*

*We want to warmly thank both of You for your thorough reviews of this paper and providing such excellent comments and suggestions the addressing of which has had a very positive impact on the paper.*

**Anonymous Referee #1**

This paper presents a factorisation analysis of a very long time series of ACSM data from the Hyytialla site. While PMF analysis has worked well for short AMS and ACSM datasets, multi-season and multi-annual datasets have previously necessitated breaking the dataset down into small chunks or using a 'rolling window' method. The approach here is to use a 'rolling relaxed CMB' approach, with the a priori mass spectral profiles generated using the unsupervised clustering of bootstrapped 'rolling PMF'. This is relevant for ACP, because while this paper does have a heavily technical bent, it also reports general interpretations of the driving factors behind aerosol behaviour at this heavily-studied site that may have implications for other studies. While this paper presents a very detailed account of what was done in terms of analysis, I do not find myself completely convinced that the authors have sufficiently argued the case for why this approach should be considered superior to other existing techniques.

To be clear, I am not necessarily saying that they should have done anything differently (I would particularly commend the adherence to unsupervised methods for the sake of objectivity), but my opinion is that the authors need present stronger arguments for why certain choices were made in the approach to data analysis and they also need to go further in exploring the strengths and weaknesses of this approach (compared to others) rather than simply accept the outputs at face value.

It's also not completely clear to me how different the interpretation of the results has been positively influenced by the use of the new technique. Put simply, it's not very clearly spelled out what the authors were trying to achieve with this approach, what the underlying principles were (as opposed to the detail of the algorithms), and whether they objectively succeeded in meeting their original objectives. For these reasons, I am recommending publication subject to major revisions. I would ask the authors to consider the points below.

**Major comments:**

It took me several reads before I really thought I understood the philosophy behind this technique, as there are a frankly baffling number of analysis stages. The authors need to be much clearer in how they describe the approach and I don't think presenting it in a stepwise form in figure 1 really helps. Instead, it should be made clear from the beginning that (assuming I did understand it correctly) the ultimate goal is to generate factorisation outputs using rolling rCMB and the biggest emphasis should be placed on this technique. As far as I can tell, all of the other stages leading up to this are merely generating objective mass spectral profiles that this algorithm uses as an input.

*Response: The reviewer is right that the big picture behind the newly tested methodology was somewhat hidden behind the large number of detailed steps presented. To overcome this issue, we have now*

   *1. made clarifications to the manuscript abstract (L16—L25; see changes below),*

2. *designed a more general process flow chart to serve as Figure 1 in the manuscript, and moved the previous Figure 1 to Appendix A (Fig. A.1)*
3. *made clarifications also to Fig. A.1 caption (note additional references to manuscript subsections as suggested by Anonymous Referee #2).*
4. *made clarifications to conclusions.*

***Changes in the manuscript:***

- **L16–L25:** "Similarly to other, previously reported efforts in OA source apportionment from multi-seasonal or –annual data sets, we approached the OA characterization challenge through Positive Matrix Factorization (PMF) using a rolling window approach. However, the existing methods for extracting minor OA components were found to be insufficient for our rather remote site. To overcome this issue, we tested a new statistical analysis framework. This included unsupervised feature extraction and classification stages to explore a large number of unconstrained Positive Matrix Factorisation (PMF) runs conducted on the measured OA mass spectra. Anchored by these results, we finally constructed a relaxed Chemical Mass Balance (CMB) run that resolved different OA species from our observations. The presented combination of statistical tools provided a data driven analysis methodology, which in our case achieved robust solutions with minimal subjectivity."

- **Figure 1 + caption:**

[Figure]

"**Figure 1** A pyramid flow chart roughly describing the steps in this work. Each block builds on the one below it, in this case starting from data collection, and ending at the final OA model, i.e. the time series of OA sub-species making up the total OA signal. The statistical analysis steps (in green) are explained in sections 4 and 5 as well as listed in Appendix A.

- **Figure A.1 caption, L1260—L1267:** "Work flow describing the machine learning analysis approach utilized in the current study. In a nutshell, this method describes how K-Means clustering can be used to classify OA mass spectral profiles from a large number of unconstrained rolling PMF runs and how this information can be further utilised to construct a data driven, relaxed CMB model to resolve the OA classes' temporal behaviours. The method comprises four main phases:  1. Performing rolling PMF (Sect. 5.1), 2. Performing window-by-window (file-by-file) clustering of rolling window iterations

(Phase I clustering; Sect. 5.2.1), 3. Conducting overall classification of the centroids calculated for all PMF windows (Phase II clustering; Sect. 5.2.2), and finally 4. Performing rolling relaxed- chemical mass balancing using the centroids retrieved in the previous step as relaxed CMB anchors (Sect. 5.3). Sections 4 and 5 in the paper introduce all the vocabulary needed for understanding this figure. These sections also contain detailed descriptions of each step in the method."

- **L893—L904:** "The rolling PMF yielded an extremely large number of PMF solutions (20 900 solutions, 62 700 factor profiles). We explored the PMF profiles across the solution space using K-Means clustering to gain understanding of the dominant OA types at the station. We revealed/identified three significantly different OA clusters: low-volatility oxygenated OA (LV-OOA), semi-volatile oxygenated OA (SV-OOA) and primary OA (POA) from these data. To attain their temporal variabilities, we performed a rolling relaxed Chemical Mass Balance (rolling rCMB) run, anchored by the observed clusters and their intra-cluster variabilities as opposed to the more conventional methods introduced e.g. by Canonaco et al. (2021). The selection of K-Means and rolling rCMB combination instead of a conventional rolling PMF enabled us to quantify POA at SMEAR II. The rCMB run explained ca. 70% of the observed OA at SMEAR II and nearly two thirds of the unexplained variation was due to high noise level of the data leaving the real unexplained variation at only 11%. The analysis method utilized here turned out to be robust, and it required little analyst interference. Therefore, our framework presents a technique to effectively analyse long-term AMS or ACSM datasets while reducing subjective bias upon analysis. However, more work is potentially needed in the future to optimize the analysis stages proposed."

The use of clustering represents the part of the analysis that is traditionally used least ACSM analysis and therefore represents the part with the least precedent. Certain decisions are made at various stages of the clustering and while the authors present arguments for why these would be considered reasonable, they do not really explore the notion of what would happen if they had done it in a different manner.

*Response: While the tested methodology works well for our data set (is clearly superior to existing techniques at SMEAR II as the existing techniques would not reveal the presence of POA), more work is needed to prove that the method presented would perform well or be superior to existing techniques in other environments. We acknowledge that in the previous version of the manuscript we came a bit strong introducing the new methodology as optimised. Therefore, sentences such as "we developed a new methodology" have been replaced with "we tested a new methodology" to make it clear that there is certainly room for further optimisation if others approaching source apportionment following our ideas and suggestions find it necessary.*

*We wish to stress that the methodology-related decisions taken (e.g. utilisation of k-means, cosine distances, weighting functions, silhouette scores etc.) yielded a fruitful rolling rCMB output as can be seen from both residual analysis point of view as well as through the unique responses of the various OA types to meteorology, which has in general been shown to be the main driver in aerosol composition fluctuations at SMEAR II (Heikkinen et al., 2020). Of course, various correlation analyses (with OA types and their known markers) could be useful, but as stated before, this approach does not work well at SMEAR II if the markers for example undergo chemical transformations at different speed compared to the OA types. What is extremely important to keep in mind is that many of these methodology-related decisions are based on Äijälä et al. (2017, 2019)*

*publications in which C-TOF-AMS PMF solutions were clustered using k-means. Many of the reviewer's comments have been addressed in Äijälä et al. (2017). Due to the similar data structure (ACSM vs C-TOF-AMS), we did not aim on replicating the Äijälä et al. (2017) findings regarding the best practises chosen for the AMS PMF factor profile classification. Importantly, we do realize that the readers need to be more clearly directed to our earlier work (i.e. the Äijälä et al. papers), and have now added the appropriate references to the text.*

*In summary, while we feel we convincingly show that our choices worked well for our data, our aim was not to suggest that any of these choices were necessarily the best ones. The above response largely covers our response also to the specific points below, but we nevertheless respond to each one with the arguments we used to choose them in the first place, hoping that this may be useful for others aiming to improve the methodology in future.*

I would specifically point to:

1. The use of k-means as a clustering algorithm.
**Response:** *This decision was made based on Äijälä et al. (2017; 2019) studies, which also utilised k-means. As stated in Äijälä et al. (2017) the combination of data dimensionality reduction (factorization) with exploratory classification (clustering with the k-means algorithm) showed that the results not only reproduced and supported earlier findings, but also (potentially) widen our current understanding on aerosol chemical composition. The decision of utilising k-means in the first place in the Äijälä et al. studies was simply to test a fast/efficient clustering algorithm that is easy to use and the clustering results are given in easy form to interpret. Due to the k-means algorithm's success in OA mass spectral classification in Äijälä et al. studies as well as within this work, we saw no incentive to perform further testing using more complex classification algorithms. However, we do understand the reviewer's concern in stating that k-means is the obvious classification tool for this purpose and perhaps if the methodology proposed here were suggested to routinely follow all the PMF analyses conducted in the future, it could be useful to explore the classification outcomes using also alternative clustering methods. We believe this could serve as an interesting follow-up study.*

*For the record, we did perform tests also with fuzzy c-means clustering (Bezdek, 1981) within this very project thinking it would better capture a continuum between fresh and aged SOA types. However, the fuzzy split between clusters complicated the further utilisation of the cluster centroids and we did not pursue this path further.*

***Changes in the manuscript:***
- **L402—L404:** "K-Means was selected as the clustering algorithm due to previous successful OA mass spectral classification performed by Äijälä et al. (2017, 2019) using this method. Future work could be conducted in exploring the potential of other clustering algorithms."

2. The use of cosine angle rather than Euclidian distance as the distance measure.

*Response: As the MATLAB k-means algorithm enables the incorporation of distance metrics other than the squared Euclidean distance, Äijälä et al. (2017) explored the various metrics available and the effects on the clustering result were mapped in Äijälä et al. (2017) Figure S.7. What can be seen from this figure is that the information value (highest number of distinct OA profiles, with structures supported by literature) was highest when utilising cosine or correlation as distance measures compared to others. As both cosine and correlation were tested in this work and they showed no evident differences in the clustering outcome, we decided to report the results gained using cosine angles due to the metric's applicability in measuring mass spectral (dis)similarities (e.g. Stein and Scott, 1994; Isokääntä et al., 2020) while being theoretically less sensitive than the correlation metric to a small number of dominant, high signals. The brief clustering tests conducted with squared Euclidean distances in this work resulted in resolving a POA cluster with a weaker structure (silhouette = 0.11±0.07) compared to the presented clustering outcomes with cosine angles (silhouette = 0.35±0.17).*

*Changes in the manuscript:*

- **L416—L424:** "Importantly, the following clustering of bootstrap iterations one rolling window at a time was conducted using cosine (dis)similarity (Sokal and Sneath, 1963) as the *k*-means distance metric as opposed to the commonly used squared Euclidean distance. This decision was again based on our earlier work in which various *k*-means distance metric alternatives were explored, and best classification outcomes (i.e. highest number of mathematically well-structured clusters, the centroids of which resembled well-known OA types found in the literature) resulted from clustering efforts utilising cosine angles along with correlations (Äijälä et al., 2017). While nearly equally good clustering outcomes were achieved between these two metrics, we decided to report the cosine (dis)similarity results due to the popularity of cosine angles in mass spectral comparisons (Stein and Scott, 1994)."

3. The weighting of higher mass to charge ratios.

*Response: The dynamic weighting of m/Q (Eq. 6 in the manuscript) was done based on our previous recommendations for K-Means clustering of Aerosol Mass Spectrometer mass spectra (Äijälä et al., 2017), where we followed work of Stein and Scott (1994). We saw no need to replicate our previous tests. However, it is important the readers are better directed to our previous study.*

*Changes in the manuscript:*

- **L411—L412:** "We previously showed the information value gains of mass scaling in conjunction with clustering AMS data (Äijälä et al., 2017). Indeed, if not applied, several OA types could not be classified (Äijälä et al., 2017)."

4. The choice of methods used to determine the optimum number of clusters and centroid profiles.

*Response: The optimal number of clusters/centroid profiles was determined using silhouette values. Four diagnostic metrics, provided within the MATLAB statistics toolbox, were tested in our earlier work (Äijälä et al., 2017). Importantly, only silhouette values enabled incorporation of distance measures other than the sqEuclidean. Due to the decision of utilising other distance metrics than sqEuclidean, we chose also to utilise silhouette scores.*

*Changes in the manuscript:*

- **L428—L430:** "Silhouette values were utilised to evaluate the clustering outcome similarly to Äijälä et al. (2017). Other metrics were not tested within this work as they would operate only by using squared Euclidean distance measures within our analysis software, MATLAB 2017a."

While I am not questioning the individual choices, I would expect the authors to offer more reasons why alternatives were discounted, reporting on any undesirable behaviour when alternatives were test, where available. I would consider the benchmark comparison of this technique to be rolling PMF, as this is an already-developed technique that is used to interpret long ACSM datasets.

*Response: It should be noted that the traditional rolling PMF does not enable us to reliably extract information regarding POA at SMEAR II. Constraining HOA or BBOA does not work with a traditional a-value approach (constraints cannot be verified through correlation analyses) and their contributions tend to be highly overestimated. For example, having PMF outputs suggesting OA mass fractions of 30% for BBOA and 10% for HOA throughout the year can be easily achieved if a-values of 0.3 and 0.1 for BBOA and HOA, respectively, are utilized, as suggested by Crippa et al., (2014). The challenges in traditional a-value approach is now highlighted in Sect. 5.2 in the manuscript. What we see from e.g. Fig. S.6 is that POA is present rarely at OA mass fractions exceeding 6%. These high overestimations of both POA types already speak for the fact that constraining them is not justified, at least for the whole data set with the chosen a-values, and due to the lack of appropriate markers, we could not find time periods when it would be appropriate to constrain them neither. Furthermore, as the results from unconstrained PMF show, HOA and BBOA do not appear in PMF in as such which complicates the task even more. Due to these reasons we cannot provide any proper comparison with rolling PMF unless performing the comparison among just the two main OOA factors, which we find to be off-topic of this study's aims. The added value of this methodology is incorporating the data-driven, justifiable, and in our view much more objective modelling of minor aerosol classes in rolling PMF. Our machine-learning-inspired methodology thus allows us to more fully exploit the strengths and minimise the weaknesses of the high data amounts of traditional rolling PMF.*

*What might interest the reviewer is that we have performed a ca. year-long rolling PMF in conjunction with a large European collaboration (referred here to as Euro rolling PMF) in which the POA profile among with its intra-cluster variability attained from the classification performed in this work was constrained and the two OOA factors were set as free. Notably this was done also with a narrower rolling window width and shift (14 days and 2 days, respectively). The figure below (Fig. AR.1) shows cumulative distribution functions of the POA mass fractions from the Euro rolling PMF and rolling rCMB, respectively, when POA was **not** found in the free rolling PMF conducted for this study. The key difference is that Euro rolling PMF predicts much higher POA concentrations for time periods when POA was not deconvolved with the unconstrained PMF. This hints towards POA overestimation in the Euro rolling PMF. We have not explored the reasons behind this phenomenon further.*

[Figure]

[Figure]

**Figure AR.1 (Left)** A cumulative distribution function of the Euro rolling PMF POA mass fraction when POA did not appear in unconstrained PMF. **(Right)** A cumulative distribution function of the rolling rCMB POA mass fraction when POA did not appear in unconstrained PMF. The green dot represents the 95th percentile, which corresponds to a POA mass fraction of 30%. Note the difference to Fig. S.7 in which the median POA mass fractions per rolling window are visualised.

*Importantly, during time periods when POA was picked up by the initial unconstrained rolling PMF, both Euro rolling PMF and rolling rCMB pick up the signal very nicely as shown in the time series below (Fig. AR. 2) while the Pearson correlation coefficient between the time series is 0.62 (p=0). The agreement between the SV-OOA and LV-OOA time series among Euro rolling PMF and rolling rCMB, which were sorted based on $f_{44}$ and $f_{43}$ ratios, is excellent (R = 0.99 for LV-OOA, R=0.95 for SV-OOA, p=0 for both).*

[Figure]

**Figure AR.2** Comparison of POA time series during an event upon which POA was found with the initial unconstrained rolling PMF.

*We feel including these comparisons in the manuscript might complicate the paper even more by direct attention to tangential questions not in the focus of this paper, such as why does the Euro rolling PMF overestimate POA.*

***Changes in the manuscript:***

- **L385—L405: "**Selecting and sorting the rolling PMF output via various criteria into three factors would have required a significant understanding of the PMF output beforehand. Choosing solid criteria can be straightforward near known pollution sources, but in case of multiple unknown factors and distant sources such becomes complicated. SMEAR II represents a station with minimal anthropogenic sources. To exemplify the challenges in correlation-based criteria at SMEAR II, we can take the correlation between NOx and HOA as an example. Both of these species are emitted from traffic and known to correlate well near traffic sources. However, in the case of transported traffic emissions, many things can affect the life time of the emitted species, which affects the correlation between the emissions at SMEAR II. If we pick the effect of wet deposition as an example, it will remove the

particulate HOA much more efficiently than gaseous NOx. If HOA and BBOA were constrained within a SMEAR II OA PMF run, it would not be surprising that the PMF output would suggest that 10% of the OA mass was made up of HOA and 30% of BBOA. As shown later on in this paper, these number are highly unrealistic. Due to the difficulty in interpreting correlations between HOA and BBOA and their markers, correlation analyses do not directly answer when constraining HOA or BBOA would have been appropriate. This is why traditional rolling PMF techniques would prevent us from HOA/BBOA quantification. This complexity motivated us to 1. use mass spectral clustering to explore the types of OA resolved within the unconstrained rolling PMF runs (i.e. answering when HOA/BBOA were present) and 2. performing rolling rCMB (Sect. 5.3) to explore the temporal behaviour of these OA types. The clustering-based exploration of the unconstrained PMF profiles was conducted PMF window-by-window across various bootstrapped PMF iterations (Phase I; See detailed description in Sec. 5.2.1). This step was followed by exploring the number of clusters across all PMF windows by further clustering all the Phase I cluster centroids (Phase II; See detailed description in Sec. 5.2.2). All the clustering procedures conducted in this study were performed within MATLAB 2017a using the *kmeans* algorithm, which utilizes K-Means++. K-Means was selected as the clustering algorithm due to previous successful OA mass spectral classification performed by Äijälä et al. (2017, 2019). Future work could be conducted in exploring the potential of other clustering algorithms."

In figure 4, the authors compare weighted residuals and the reader could be forgiven for thinking that this has resulted in an inferior data product to what was obtained at the end of stage 1 of the analysis.

***Response:*** *This is a valid point and the comparison of the initial PMF $Q/Q_{exp}$ vs the rolling rCMB $Q/Q_{exp}$ could be confusing. The point of Figure 4a is that while PMF residuals always increase when PMF constraints are applied (e.g. Canonaco et al., 2013), the observed increase is small, which for example was not the case in the Canonaco et al. (2013) CMB exercise (note that this CMB was not relaxed) (personal communication with F. Canonaco, 2021).*

***Changes in the manuscript:***

- ***Figure 4a:*** *Figure is moved to the supplementary material (now Fig. S.3)*
- ***L516—L523:*** "To evaluate the averaged rolling rCMB output, we first compared the Q/Qexp values between the initial rolling PMF and rolling rCMB. The comparison of the Q/Qexp retrieved from each iteration in each rolling window is visualized in Fig. S.3. As expected, the mean rolling rCMB Q/Qexp value was higher (38% increase) than that of the initial rolling PMF Q/Qexp. This is typical as Q/Qexp tends to increase whenever constraints are added to the PMF run. However due to the relaxed approach, the Q/Qexp increase is for example much less dramatic than shown in Canonaco et al. (2013) CMB tests. We find the observed Q/Qexp increase acceptable, considering the higher information value (interpretability) provided by the rCMB solution."

The authors need to discuss in more depth the pros and cons of each approach and present a stronger case for why it should be considered advantageous to use the technique used here.

***Response:*** *The main strengths of the technique introduced here was that it*

1. *enables us to quantify POA, which the traditional a-value approach would not be able to do due to above-mentioned challenges in correlation analyses*
2. *the method could reduce user bias and could be less laborious than the traditional approaches.*

*We believe that both points are clearly stated in the manuscript, and stated also both in the abstract and conclusions (see answers above).*

This should bear in mind that just because a particular algorithm uses less supervision and produces a less ambiguous result, these do not in themselves mean the data products are intrinsically more accurate. As part of any numerical data reduction such as this, it is vital to properly explore its limitations and I don't consider what is presented here to be sufficient. I am particularly interested in the analysis of residuals, which is normally the first thing to inspect. Is the unexplained variance and mass shown in figure 8 purely random noise, or is there any structure (relative to m/z, season, temperature, time of day, etc.) that might suggest there are factors at work that this does not adequately capture?

***Response:*** *To provide a more comprehensive look into residuals we had a closer look in the scaled residuals as well as the unexplained variance that was not related to noisy signals.*

*First, as an exercise we performed k-means clustering in a similar manner to Phases I and II on the scaled residual matrix. The figure below (Fig. AR.3) reveals the lack of mass spectral structures in the residual matrix. However, if taking a mean of the negative scaled residuals alone, a resemblance with the POA mass spectrum can be seen (Fig. AR.4) as discussed already in the previous versions of the manuscript. As discussed in the manuscript, this potential overestimation (shown primarily in summer) was still extremely minor as can be seen from the POA OA mass fractions in manuscript Figure 8.*

[Figure]

**Figure AR.3** Median silhouette scores obtained from k-means clustering (using cosine angles as distance measures) on rolling rCMB the ***scaled residual matrix.*** The scaled residual was mass scaled similarly as done within Phase I and II clustering in the tested methodology.

[Figure]

**Figure AR.4** In green: median taken over all the *negative* scaled residual matrix time steps for each *m/Q*, respectively. In blue: POA spectrum.

*Due to questions raised by anonymous reviewer #2 we also explored the effect of downweighting on the PMF results. The key differences observed were related to the scaled residual histograms (see Fig 4 below). When downweighting was applied using the cell-wise with a $(SNR)^{-1}$ function, we observed negative, near-zero spike in the histogram (Fig. 4 green trace) while the rolling rCMB scaled residual histograms without downweighting was Gaussian (Fig. 4 purple trace). The slightly negative near-zero scaled residuals appeared mostly in summer (Fig. S.4a) and could be related to time periods with high signal-to-noise (SNR; Fig. S.4b). Slightly similar SNR-dependence of scaled residuals could be observed among scaled residuals if no downweighting was applied, which could hint towards non-optimal determination of errors during the high signal periods. However, we do not find these features alarming as the residuals are extremely small and range between ±4. For example, the recently published rolling PMF results for Zürich have scaled residuals ranging between ±10 (Canonaco et al., 2021).*

*We also present the annual median scaled residual spectra as well as time series for each measurement year respectively so the readers have the possibility to see that no evident long-term trends exist and the the residuals are primarily just noise (Fig. S.5).*

*Finally, we show that the (real) unexplained variance has no seasonal or diurnal cycles (Fig. AR.5a) and no correlations can be found between temperature, total organic aerosol (Org), sulphate (SO₄), nitrate (NO₃) or equivalent black carbon (eBC) (Fig. AR.5b). To conclude, no evidence coulb be found from the residuals that would suggest poor/unacceptable performance of rolling rCMB that would make us question any of the rolling rCMB outputs.*

[Figure]

**Figure AR.5** (a) The median diurnal cycles for each month of real unexplained variance obtained from rolling rCMB. The y-axis represents the time of the day and x-axis month of the year. The color code reflects the real unexplained variance which is expressed in percentages. (b) A correlation matrix (Pearson correlation on the left column and the corresponding p-values on the right column) between the real unexplained variance and temperature, total OA, sulphate, nitrate and equivalent black carbon.

*Changes in the manuscript:*

- **L515:** 5.3.1 Rolling rCMB residual analysis and output evaluation
- **L528—L547:** "The scaled residual histogram, presented in figure 4 in green, is fairly unimodal and spreads between [-4, 4] (most data between [-3, 3]) as desired (Paatero and Hopke, 2003), but tends to have high frequency of slightly negative, near-zero readings. We connected this behaviour to periods with high SNR (i.e. summers; Fig, S.4). As downweighting of the noisy and weak variables made as a function of SNR-1 (Sect. 4.1.1 and 5.1) which further influences $\sigma_{ij}$ in Eq. (8), the seasonality in SNR was seemingly driving the scaled residual seasonal cycles. This was visible, yet to a lesser extent, in a test rCMB run conducted without downweighting (Fig. 4 in purple) and with a more traditional average step wise downweighting procedure (not shown), which further brings us to the conclusion that the PMF input matrix errors are also SNR-dependent (Ulbrich et al., 2009.) and could perhaps be further optimised. However, it should be kept in mind that the scaled residuals in general speak for a good performance of rolling rCMB in modelling the input data, and the scaled residual time series shown in Fig. S.5 reveal no evident patterns/trends except the negative values in summers. An additional investigation into the real unexplained variation within the data (shown later on in Fig. 8) revealed no correlations with temperature or sub-micrometre PM components.

  Annual median scaled residual mass spectra are visualised in Fig. S.5. Even clustering attempts on the scaled residual matrix do not reveal clear structures in the scaled residual matrix although an overall median scaled residual mass spectrum calculated using the negative residuals alone would hint towards some resemblance with POA at $m/Q > 50$ Th. We note that this could indicate minor POA overestimation in the rolling rCMB and speculate whether introducing time-dependent profile variation limits to ME-2 could help us overcome the issue. With the method presented here, we could easily extract time-dependent limits for ME-2 variability. However, introducing such to dynamic approach to the ME-2/SoFi Pro analysis software is not yet possible."

- **Figure 4:**

[Figure]

**Figure 4** Normalized histograms (probability density function, PDF) of the scaled residuals obtained from rolling rCMB. The effect of downweighting weak/bad variables is visible by the high scaled residual frequencies at negative near-zero readings. If rolling rCMB was conducted without downweigting the scaled residual distribution behaves in a highly normal manner.

[Figure]

**Figure S.4** The seasonal diurnal cycles of the scaled residual is visualised in panel (a) and 3d histograms of the scaled residual as a function of signal-to-noise (SNR) is presented in panels (b) and (c). The difference of panels (b) and (c) is that the weak variables were downweighted cell-wise using the (SNR)-1 method (Sec. 4.1.1) for panel (b) and panel (c) holds data from an rCMB run without downweighting bad or weak variables. As shown in panel (a) no clear diurnal patterns in the scaled residuals can be observed, but a seasonal cycle exists. As reflected in panels (b) and (c), this seasonality can be explained by the mean SNR, which is highest in summer when the OA mass loading is at highest (e.g. Heikkinen et al., 2020). The relationship between the scaled residual and SNR is highest when the errors have been further downweighted for rCMB to reduce the weight of weak and noisy variables within the PMF iterations.

[Figure]

**Figure S.5** The annual median (IQR shown with error bars) scaled residual spectra (left) and time series (90th and 10th percentiles with light grey shadings and IQR with darker grey shadings). Both show primarily noisy features. The slightly negative scaled residuals shown in figure 4 are visible during summers. These periods are associated with high SNR (i.e. summertime at SMEAR II).

A key detail in the final data products is by how much the mass spectral profiles of the factors was allowed to vary in the rCMB analysis. As I understand it, if the profile could vary with the rolling window, then this would avoid many of the limitations imposed by the default PMF data model, however this then creates implications for how the results are interpreted, in particular with the seasonal analysis and long term trends. Can the authors be sure that any interpretations presented in section 6.2 are the result in changes in the abundance of the different organic aerosol types, or changes in the mass spectral profile?

*Response: The trend analysis could have been impacted by the amount which factors were allowed to vary. Fig. AR.6 shows the annual median $f_{44}$ and $f_{43}$ values for LV-OOA and SV-OOA, respectively, along with their interquartile ranges with error bars. $f_{43}$ seems to be lower in 2012 compared to other years for both OA types and could therefore indicate that the OOAs were less efficiently modelled in 2012 compared to other years. We have, however, decided to exclude the trend analysis from the manuscript due to multiple issues that complicate the analysis (gaps in data and measurements conducted without a dryer in the beginning). Other than 2012 being slightly off, the clusters are dense as also shown with the cluster silhouette scores e.g. in Fig. 5 titles and the strong agreement between the OOAs when comparing the Euro PMF to rolling rCMB. No strong evidence exists to think that the interpretations in section 6.2. would be introduced by the methodology itself.*

*However, we do believe that introducing time-dependent limits to ME-2 would further improve the rCMB result as discussed in Sect. 5.3.1.*

[Figure]

**Figure AR.6** Annual IQR of $f_{44}$ vs $f_{43}$ visualised for LV-OOA and SV-OOA clusters, respectively. As hinted by the median cluster silhouette scores, the $f_{44}:f_{43}$ are very similar across the clusters.

If, on the other hand, the profiles are rigid, then this instantly asks questions of whether the factorisation is equally applicable at all times, or whether the technique is still susceptible to the same rotational ambiguity problems as conventional PMF. Generally speaking, the explanations of the behaviour in section 6.2 are largely speculative and seem to be focused on creating a plausible narrative rather than offering new scientific insight. The authors should focus the discussion on what this new work adds to the (already substantial) body of work concerning this site and tropospheric aerosol processes in general, specifically through the virtue of adopting this new technique (as opposed to other existing methods). While hypotheses are frequently referred to casually in the text, it's not completely clear how these are being tested by the data and to what certainty.

***Response:*** *Rotational ambiguity (RA) in factor analysis is indeed an inherent limitation and challenge of complex ambient air data analysis especially, and complicates the selection of PMF/ME-2 solutions and their interpretation. The additional analysis layer we have in our methodology is designed specifically to help the analyst decide which of the RA behave stably over time and thus meaningful in the analysis. However, we can not follow the referees thinking here: what does "whether the factorisation is equally applicable" refer to here? In our understanding, constrains in factorisation reduce the amount of allowed rotations (i.e. reduce RA). So RA of ME-2 or rCMB should generally be less of an issue than in free PMF. If anything, we believe the along number of solutions in our first, the proneness of free rolling PMF to RA in the first, exploratory stage may actually help in bringing to the table the numerically rarer, but realistic and chemically important, POA containing solutions in addition to the starting seed-wise more common main OOA splits. The subsequent stages of our method then suppress the rotational outliers that did not appear consistently. However, important as it is, RA is far from the only measure of applicability of factor analysis, and e.g. solution quality (here $Q/Q_{exp}$, residuals) and interpretability can be considered equally if not more important. We thus would see RA not be a major concern in the applicability of our method, and, as covered in the previous answers, we find we can similarly define and justify the rigidities and variability limits in a superior way to the traditional rolling ME-2 a-value and correlation studies.*

*We would also like to stress that **long-term observations** of OA composition have never been reported from this site and thus no substantial body of work exists regarding this very topic. The aim of this paper is to provide*

*first insights into the OA climatology at SMEAR II. The distinct responses of OA composition to changing meteorology among the three factors do speak for atmospheric relevance of the rolling rCMB solution. It should be noted that meteorology is the main driver causing variability in PM composition at SMEAR II due to the lack of anthropogenic emissions at the site and the strong meteorology-dependences on air pollutant transport and biogenic SOA production. This we show very clearly in the manuscript.*

**Minor comments:**

The term 'openair' is used to identify certain types of plots, but openair represents a large suite of many different graphing tools. Furthermore, openair wasn't actually used to generate the ones here. While it may be appropriate to credit the development of openair with popularising these graph types, the plots should be referred to by their specific type, e.g. polar plots.

*Response: ZeFir is actually using the Openair R-package in the background for the calculations. For drawing these figures ZeFir acts only as an alternative graphical user interface. We find polar plot too general and non-descriptive of the actual content of the data displayed as polar plot could refer to anything drawn using polar coordinates. We used Openair polar plot terminology in our previous publication describing SMEAR II aerosol chemical composition (Heikkinen et al., 2020). Not to confuse readers to think the plots represent different things in these two related papers, we keep referring to the figures as openair polar plots for the sake of consistency. However, we will change the 6.3.1 title to a more informative form.*

***Changes in the manuscript:***

- **L795:** 6.3.1 Wind direction and speed dependency of OA composition

I found section 6 overly wordy, with more text than was necessary to convey the important information. The authors may wish to cut back on the amount of discussion presented concerning the approach to analysis, instead focusing on conveying the new scientific insights this offers, ideally in the form of hypothesis testing. One of the problems that has traditionally confounded long term PMF analysis is that the mass spectral profiles of OOA factors can vary with season. Did the authors find any evidence of this at any stage of the analysis?

*Response: As shown in Fig. 3f, the silhouette scores for SV-OOA expressed seasonality hinting of seasonal variability in SV-OOA mass spectra. However, the high silhouette scores implies that this seasonality is not significant enough to split SV-OOA into two separate clusters and the analysis of the spectral changes within the SV-OOA profiles are not obvious.*

> *Section 6 is shortened due to the removal of the long-term trends subsection.*

Regarding figure 6, the overlaid ellipses are not necessary, given that the points are already coloured. I would remove them, as they only serve to distract.

*Changes in the manuscript:*

- **Figure 6:**

[Figure]

**Figure 6** (a) A triangle plot visualizing the mass spectra distribution in each cluster in $f_{44}$ vs $f_{43}$ space, (b) Van Krevelen diagram visualizing the mass spectra in H:C vs O:C space for LV-OOA and SV-OOA, (c) mass spectra in $f_{44}$ vs $f_{60}$ space for indications of fresh BBOA, (d) $f_{55} \times f_{57}$ vs $f_{60}$ space for indications of HOA and BBOA.

Also regarding figure 6, a number of points associated with LV-OOA have a high $f_{60}$, which is a classic symptom of LV-OOA 'mixing' with BBOA, owing to the high HULIS content of the latter. The authors should comment on this.

*Changes in the manuscript:*

- **L639—L640:** "Owing to their high $f_{60}$, these outlier spectra likely originate from biomass burning, but are mixed within the LV-OOA cluster due to the high humic-like substance content of the BBOA (e.g. Ng et al., 2010)."

Section 6.2.4: A statistical treatment is referred to, but no actual quantitative results are presented. This should be done, even if it is to report that no significant trend was observed.

*Response/Changes in the manuscript: After a careful consideration and comments from reviewer #2, we ended up removing the discussion on long-term trends due to the high uncertainty in the trend analyses.*

**Anonymous Referee #2**

General comments: This manuscript describes an interesting approach to deal with the source apportionment of long submicron organic aerosol (OA) datasets at a remote site (SMEAR II research supersite in the boreal forest). At these types of sites, far from primary sources, a "classical" rolling PMF method (commonly based on the identification and constraining of primary factors through a criteria-based approach) may not be sufficient to deal with mostly only secondary factors.

This article presents a new methodological approach based on K-means clustering of a very large number of unconstrained rolling PMF solutions, to overcome this difficulty. The extracted clusters are further used to run a CMB-type (constrained) rolling PMF. Output results are finally utilized to provide an overview of the temporal variability of the three OA factors identified. Methodological results (section 5) extend from pages 11 to 21 (with about 3 pages of Figures), while the "Results" section (section 6) ranges from page 21 to 37 (with about 7 pages of Figures). The general impression is that the paper permanently hesitates (and authors have been unable to choose) between the two focal points (methodology- or result-oriented) which is confusing and frustrating at times, especially when the methodological choices seem too descriptive (see specific comments below). The overall quality of the manuscript is excellent. A few minor technical corrections are reported at the end of this document.

*Response: We recognize this hesitation. The "issue" is that the analysis of the data using a new set of analysis tools required detailed description. This detailed description is required to increase confidence regarding the results obtained with a new methodology as well as a chance for others to analyse their data sets in a similar manner and perhaps further optimise the steps we suggested. Although it was considered, we decided not to separate the methods and the atmospheric results into two papers, as much of the validation of the methodology comes from the consistency and quality of the final results. Where possible, we have tried to keep sections better focused when updating them.*

**Specific comments:**

L40-41: "it is also possible that ACSM was less efficiently capturing short term (POA) pollution plumes". I believe this assumption derives from the behavior observed with the 3-hour averaged dataset. Therefore could it rather arise from the averaging (diluting the information over a longer time window) compared to the ACSM raw time resolution (and even more if compared to high time resolution AMS measurements performed at the same site) rather than from sampling differences like "capturing" may suggest?

*Response: The reviewer's suggestion is good and this point was actually stressed more in the conclusions section of the manuscript (**L924–L927**: "This discrepancy could be for example linked to a decrease in POA emissions as hinted by decreasing BC trends at the site (Luoma et al., 2020), or the ACSM limited capability in detecting short-term (pollution) plumes, which average out even more due to the 3-hour averaging applied to the PMF input data, which was necessary to improve the SNR at this rural background site"). We modified the manuscript abstract also accordingly.*

***Changes in the manuscript:***

- **L44–L47:** *"While the co-located long-term measurements of black carbon supported the hypothesis of higher POA loadings prior to year 2012, it is also possible that short term (POA) pollution plumes were averaged out due to the slow time resolution of the ACSM combined with the further 3-hour data averaging needed to ensure good signal-to-noise ratios (SNR)."*

L171 & L174: "Open air (...) analyses" and "Openair polar plots" are not informative titles and should be modified. Maybe something like "Influence of meteorological parameters on air pollutant concentrations" for section 3, and "OA variation with wind direction and speed" for subsection 3.1 (or something along that line)? See also section 6.3.1 (L789).

*Changes in the manuscript:*
- **L794:** *6.3.1 Wind direction and speed dependency of OA composition*

L183: In the back trajectory and TOL analysis, were the possible rainfalls along the airmass trajectory taken into account since it could highly influence the aerosol loading arriving at the site?

*Response: It is true that cloud processing in general and subsequent precipitation will influence aerosol size distribution during the transport to the observation site. For example, aerosol activation to cloud condensation nuclei and formation of cloud will initiate liquid phase oxidation of SO$_2$ into sulphate aerosol. If the cloud does not precipitate but evaporates, the dried aerosol size distribution will feature a Hoppel minimum at the edge of the activation size. Consequently the chemical composition will be predominantly non-hygroscopic on the non-activated Aitken sizes and more hygroscopic due to sulphate aerosol at the accumulation mode sizes. If the cloud formation leads to precipitation, this will reduce the aerosol concentration overall and enhance wet scavenging processes. Overall, aerosol-cloud-precipitation processes will take place in a dynamic manner during the transport. There is definitely a need to explore these features in a systematic manner in the future. However, in this study we did not take these interactions and precipitation processes into account. Our aim was to explore the net effect of time-over-land in sub-micron aerosol chemistry at a fixed site.*

*Changes in the manuscript:*
- **L860—L864:** *"Additionally, also cloud processing and subsequent precipitation will influence aerosol size distribution during the transport to the observation site. However, in this study we did not take these interactions and precipitation processes into account. Our aim was to explore the net effect of TOL in sub-micron aerosol chemistry at a fixed site. Therefore a need to explore these features in a systematic manner in the future also exists."*

L323: Considering this is a new method, this section should not only be descriptive, but also provide some elements to compare between unconstrained rolling PMF/constrained rolling PMF/machine-learning approach ... and highlight the advantages of this latter methodology – at least in this case – to provide a more accurate analysis of the OA fraction. It is not so obvious to me after reading the manuscript what logic was behind the different validation steps (which are not so explicit).

*Response/Changes in the manuscript: We thank the reviewer for pointing this out. This comment is highly similar to those given by the Anonymous Reviewer #1. We therefore guide the reviewer to read those reviewer*

*comments and our responses as we believe to have already answered these concerns earlier and modified the manuscript accordingly.*

L330: Since a correct assessment of the error matrix is critical for PMF analysis, please describe (at least in the SI) how the error propagation was performed for the 3-hour averaged OA.

***Changes in the manuscript:***

- **L344—L345:** "The error propagation was accounted for during averaging (linear terms of the squared Taylor series expansion on the measurement data)."

L332: The most common approach uses a step function to downweight the weak and bad variables, based on averaged values for each m/Q. Although I agree with the authors that the cell-wise function may be more appropriate, especially if there is some strong seasonal variability for specific m/Q, have you tested both types of weighing to estimate the difference?

***Response:*** *Due to the reviewer's comment we ran rolling rCMB with three weak/bad variable downweighting scenarios: i) cell-wise + $(SNR)^{-1}$, ii) no downweighting, iii) averaged + step function. No major difference in the rCMB outcome was observed; see figure below (Fig. AR.7) depicting the time series comparisons between the two downweighted approaches. The major difference was observed in the shape of the scaled residual histogram as depicted in Fig. 3 and discussed together with the replies to Anonymous referee #1 (see answers above).*

[Figure]

**Figure AR.7** Comparison of LV-OOA, SV-OOA and POA concentrations obtained via rolling rCMB using two different downweighting methods: commonly used downweighting based on average signal-to-noise ratios (SNR) of PMF variables and a stepwise downweighting procedure (*y*-axes) and the cell-wise downweighting using $SNR^{-1}$ function (*x*-axes). The time series comparison is performed via a 3d histogram in which the observation frequencies (counts) are depicted with the colour scale. The figure reveals that the concentrations obtained with the two downweighting techniques agree well.

L339: If I understand correctly, since the initial rolling PMF is unconstrained, it is possible to have three factors that vary over the seasons/years. For instance, BBOA, HOA and 1 OOA in winter then switching gradually to HOA and 2 OOAs. But what if only two factors are relevant for some runs? Did you have a way to account for that (and eventually discard those runs)? It is relatively common in PMF analyses to keep only the "best" runs and discard those who do not fit criteria. Here I am not sure to understand if that happens in that first step (I believe not) or if using the weighted cluster centroids in the second phase is indeed taking care of the possible outliers.

***Response:*** *The problem of having only two relevant factors in a run is dealt during the Phase I clustering, which works as follows:*

1. *All solutions from the 100 iterations/PMF window are clustered using 2–4 clusters (k = [2,3,4])*
2. *The silhouette values (s) are calculated for each clustering solution ($s_{k=2}$, $s_{k=3}$, $s_{k=4}$)*
3. *$s_{k=2}$, $s_{k=3}$, $s_{k=4}$ are cross-compared and the highest silhouette reveals the optimal number of clusters for the PMF window*
4. *The silhouette weighted centroid for the optimal number of clusters solution is used for Phase II clustering*

*In short, the "correct" number of clusters is thus resolved mathematically for all free rolling PMF runs (time points) and the optimal number (for this point / run) is used for the subsequent analysis. Figure 3b reveals the statistics of the optimal numbers of clusters. It tells us that most of the time (ca. 80%), two factors would have been enough, but there are scenarios where 3 or even 4 clusters were present (this is when POA showed up). The decision regarding the number of clusters, i.e. the best solution, is therefore solely based on mathematics (silhouette scores). If we focus on the common scenario, when $s_{k=2} > s_{k=3}$ & $s_{k=2} > s_{k=4}$, we save the clustering solution for k=2 clustering. In reality this means that two OOA profiles could be resolved and a third of the profiles incorporated within the clustering are some unrealistic splits of those. As these splits were not similar enough, they did not sufficiently show up as their own cluster. Importantly, as they are different from the two OOAs, they represent cluster outliers and possess therefore low silhouette scores. To calculate the silhouette-weighted centroids, misclassified spectra (s ≤ 0) are discarded and then each spectrum is multiplied by its silhouette score prior to calculating the average over all the spectra belonging to the cluster. As the outlier clusters have low silhouettes, they play a lesser role determining the cluster centroid profile.*

L359 (Figure 1): I liked the summary of all the steps in that Figure but it would be interesting to make the link with the corresponding sections in the text, either directly on the Figure or at least in the caption.

***Response:*** *We decided to move this detailed figure to the Appendix and provide a more general figure in the manuscript main text.*

***Changes in the manuscript:***

- **Figure A.1 caption:** "Work flow describing the machine learning analysis approach utilized in the current study. In a nutshell, this method describes how K-Means clustering can be used to classify OA mass spectral profiles from a large number of unconstrained rolling PMF runs and how this information can be further utilised in a relaxed CMB run to gain insight into the OA classes' temporal behaviours. The method comprises four main phases:  1. Performing rolling PMF (Sect. 5.1), 2. Performing window-by-window (file-by-file) clustering of rolling window iterations (Phase I clustering; Sect. 5.2.1), 3. Conducting overall classification of the centroids calculated for all PMF windows (Phase II clustering; Sect. 5.2.2), and finally 4. Performing rolling relaxed- chemical mass balancing using the centroids retrieved in the previous step as CMB anchors (Sect. 5.3). Sections 4 and 5 in the paper introduce all the vocabulary needed for understanding this figure. These sections also contain detailed descriptions of each step in the method."

L394: There could also be strong arguments against mass-to-charge scaling for this type of measurements. Because of their low signal-to-noise ratios, high m/Qs are often downweighted (as weak variables) in the PMF analysis, even if they can contain more information on OA sources. Have you selected only those for which the SNR was high enough in the dataset? Otherwise it seems contradictory to the previous reasoning applied to PMF. And I am wondering if it could play a role in the slight overestimation of POA in summertime when applying the rCMB.

*Response: Importantly mass scaling was not part of the PMF analysis, but part of the clustering alone. We showed previously how such pre-process of scaling helps K-Means clustering in classifying POA types (Äijälä et al., 2017). The resulting highlighted structures were shown to be chemically relevant high m/Q spectral characteristics rather than noise. We cannot unfortunately follow the reviewer's thoughts on how exactly this could relate to the potential summertime POA overestimation.*

L521 and following, and Fig. S.4: What is the proportion of "the PMF windows where POA was not classified"? The number of runs for which this factor appears compared to the two others could be indicated in Fig. S.4 as well (I guess it is N = 28, mentioned in Fig. 5e which appears later in the text?). What is puzzling me is that POA should appear more "easily" in the rCMB PMF runs when its concentration is higher, and clearly this is what will drive the slope of the linear fit in Fig. S.4c. Here the slope between the two POA factors (from clustering and rCMB) stays low (0.48) despite quite a few points at high concentrations.

*Response: POA primarily appears in relatively "clean form" in 28 windows (silhouette >0) in the initial unconstrained rolling PMF (the rest not classified) and it does indeed appear more easily in the rCMB. The low slope can be explained by the fact that at SMEAR II POA plumes coincide with high LV-OOA loadings and PMF sums these signals up (not clean POA).*

*Changes in the manuscript:*
- **Figure S.6 caption:** "Figure S.6 The slope in panel (c) is highly driven by the high POA concentrations. These concentrations were higher within the initial free PMF compared to the POA deconvolced via rCMB. We could suspect that the POA profile modelled by the initial PMF at those times slightly differs from the rest of the POA spectra, which results a difference between the POA concentration between the rCMB and initial PMF."

L650: Stronger conclusions on the sources (anthropogenic or not; local or not) of LV-OOA could be drawn from various wind data analyses (backtrajectories, NWR plots, etc.). Thus it feels strange to find them here since section 6.3 specifically deals with the wind sectors influencing the sampling site. I would suggest to shorten the discussion here and refer to section 6.3 where you should be able to give stronger conclusions.

*Response: We agree that a link to the wind/trajectory section ought to be made in Sect. 6.2.1. as it contains crucial findings supporting the conclusions made here.*

*Changes in the manuscript:*
- **L694—L695:** "More discussion on LV-OOA sources, supporting the abovementioned statements on the anthropogenic and biogenic influences on LV-OOA, is presented later in the paper in conjunction with wind and air mass trajectory analyses (Sect. 6.3)."

L713: "a larger wintertime fPOA:". Confusing. I thought the discussion had switched to POA absolute concentrations and not fPOA anymore?

*Response: This was typo and the topic was POA not $f_{POA}$. It is now corrected.*

*Changes in the manuscript:*

- **L748—L750**: " Several phenomena can explain a larger wintertime POA loading: wintertime POA dispersed in a shallower atmospheric mixing layer compared to summer, and sources of POA are possibly greater in winter due to enhanced need for residential heating and less of POA evaporation due to cold temperatures."

L725: I am not convinced by the usefulness of this section, which I feel unconnected to the rest of the story. The authors themselves state that the dataset is probably not long enough to conduct a robust trend analysis, and that their assessment is likely biased by missing data for some seasons, or technical changes in the sampling line (no dryer for the two earlier years). Besides, this is clearly indicated in the conclusion as well (L916-918).

*Response/ Changes in the manuscript: We recognise these issues as well, and decided to remove the section.*

**Technical corrections:**

L33: "However, also the nearby". Please consider revising this sentence.

*Changes in the manuscript:*

- **L35—L36:** "Two nearby sawmills also played a significant role in SV-OOA production as also exemplified by previous studies at SMEAR II."

L114: "recorded"

*Changes in the manuscript:*

- *Typo corrected*

L119: "a new framework"

*Changes in the manuscript:*

- *Corrected*

L166: Please provide the CPC model.

*Changes in the manuscript:*

- **L175:** "a TSI Condensation Particle Counter (CPC; TSI 3772) as a reference instrument."

L238-240: Unclear. Please consider revising this sentence.

*Changes in the manuscript:*

- **L250—L253:** "The decrease of $Q/Q_{exp}$ as a function of $p$ can be, to some extent, used to understand what the optimal number of factors in the solution could be. While $Q/Q_{exp}$ tends to drop as a function of $p$, the optimal $p$ is typically where the $Q/Q_{exp}$ drops stop being significant."

L297: "Reassigning all the points"

*Changes in the manuscript:*

- *Corrected*

L317: "indicate a good"

*Changes in the manuscript:*

- *Corrected*

L322-323: "a relaxed (...) PMF analysis" (or another singular term)

*Changes in the manuscript:*

- **L335—L337:** "The ultimate goal of the unconstrained PMF and K-Means clustering was to provide mass spectral profiles as a priori input for a PMF run in which all of the profiles are constrained with reported intra-cluster mass spectral variabilities. This PMF approach is therefore termed as rolling relaxed CMB, i.e. rolling rCMB."

L347: "28 days". I believe it is not hours.

*Changes in the manuscript:*

- *L362:* "(tests up to window width of 28 days)."

L347: "Only window widths"

*Changes in the manuscript:*

- *Corrected*

L367 (Figure 2): please specify "time series of 3-hour averaged OA"

*Changes in the manuscript:*

- **Figure 2 caption:** "Figure 2 (a) The 3-hour averaged time series of OA measured at SMEAR II and utilized in the current study. The y-axis represents OA mass concentration in µg m$^{-3}$ and the x-axis the time. The figure also depicts the data coverage within the eight years. The yellow shaded region represents the first two months of measurement data, which are further shown in panel b. (b) Schematic figure visualizing the rolling window approach. Now, *x*-axis spans from April 1$^{st}$ to June 1$^{st}$, 2012 and the six OA time series represent the timespans of successive rolling PMF windows. With the settings used in the current study, this two-month period would be part of six rolling PMF windows. "

L509: "such a dynamic"

*Changes in the manuscript:*

- *Corrected*

L512: remove "rolling"

*Response: The small r in rCMB refers to the word "relaxed" as defined in our earlier work (Äijälä et al., 2019). Therefore, an accurate description of the method used is rolling rCMB. We noticed that the earlier manuscript version stated "relaxed rolling CMB, rCMB" when it should have said "rolling relaxed CMB, rolling rCMB". We have modified the manuscript accordingly and highlighted the changes in yellow.*

L591-592: "we can see that the our SV-OOA O:C". Delete "the".

***Changes in the manuscript:***

- *Corrected*

L627-628: "in in". Delete one.

***Changes in the manuscript:***

- *Corrected*

L663 and other occurrences in the text: "diel cycle" would be a more appropriate term than "diurnal cycle".

***Changes in the manuscript:*** *"diurnal cycle" has been replaced by "diel cycle" in the new manuscript version and highlighted in yellow.*

L671-672: seems like a sentence fragment. Consider revising.

***Changes in the manuscript:***

- **L703—L705:** "However, we are not able to confirm whether all of the SV-OOA is of biogenic origin. This is because the nearby sawmills in Korkeakoski (ca. 7 km NE of SMEAR II; Sec. 2.1) represent significant SV-OOA sources (e.g. Äijälä et al., 2017)."

L689: "is does likely not play". Please revise.

***Changes in the manuscript:***

- **L723—L725:** "In addition, upon the development of the turbulent daytime boundary layer the SV-OOA yielded during the night does likely not play any major role in the SV-OOA loading within this daytime boundary layer."

L748: "A-OA". Do you mean "POA"?

***Response/Changes in the manuscript:*** *Yes, we certainly meant POA. This error belongs to a section (6.2.4 Long-term trends) that has been removed from the new version of the manuscript.*

L757 (Figure 7): I honestly do not see a great difference between the light and darkshadings. Could you add more contrast?

***Changes in the manuscript:***

- *Figure 7:*

[Figure]

L764: "POA" in the caption does not seem vertically aligned with the others.

*Changes in the manuscript:*

- **Figure 8:**

[Figure]

L863: "As these figures, do not indicate". Delete the comma.

*Changes in the manuscript:*

- *Corrected*

L869 (Figure 12a): IQR overlap in this plot. May be more readable with one plot per factor.

*Changes in the manuscript:*

*We appreciate the reviewer's comment but we feel the figure is readable enough as is and the IQR overlap does not give room for erroneous interpretations of this figure.*

L895: "that the need (...) to increase". Please revise.

*Changes in the manuscript:*

Wording no longer used (sentence removed).

L1225-1230: Two references not in alphabetical order.

*Changes in the manuscript: We let the copyeditor move them if found necessary. Ä and A are certainly different letters and the umlauts do not refer to an accent. Finnish/Swedish alphabet ends like this: "…, Z, Å, Ä, Ö".*

**Supplementary information**

L33: "The coloured lines represent "

*Changes in the manuscript:*

- **L46:** "The coloured lines represent linear fits"

Figure S6: Plots would be more readable if vertically expanded. Coloured and grey shaded areas are not explicit in the caption.

*Response/ Changes in the manuscript:*

*Figure excluded from the updated manuscript version.*

Figure S7: IQR overlap in these plots. They may be more readable with one plot per factor.

*Response:*

*We appreciate the reviewer's comment but we feel the figure is readable enough as is and the IQR overlap does not give room for erroneous interpretations of these figures.*

**References**

Bezdek, J.C., Pattern Recognition with Fuzzy Objective Function Algorithms, Plenum Press, New York, 1981.

Canonaco, F., Crippa, M., Slowik, J. G., Baltensperger, U., and Prévôt, A. S. H.: SoFi, an IGOR-based interface for the efficient use of the generalized multilinear engine (ME-2) for the source apportionment: ME-2 application to aerosol mass spectrometer data, Atmospheric Measurement Techniques, 6, 3649-3661, 10.5194/amt-6-3649-2013, 2013.

Canonaco, F., Tobler, A., Chen, G., Sosedova, Y., Slowik, J. G., Bozzetti, C., Daellenbach, K. R., El Haddad, I., Crippa, M., Huang, R. J., Furger, M., Baltensperger, U., and Prévôt, A. S. H.: A new method for long-term source apportionment with time-dependent factor profiles and uncertainty assessment using SoFi Pro: application to 1 year of organic aerosol data, Atmos. Meas. Tech., 14, 923-943, 10.5194/amt-14-923-2021, 2021.

Crippa, M., Canonaco, F., Lanz, V., Äijälä, M., Allan, J., Carbone, S., Capes, G., Ceburnis, D., Dall'Osto, M., Day, D., DeCarlo, P. F., Ehn, M., Eriksson, A., Freney, E., Hildebrandt Ruiz, L., Hillamo, R., Jimenez, J. L., Junninen, H., Kiendler-Scharr, A., Kortelainen, A.-M., Kulmala, M., Laaksonen, A., Mensah, A. A., Mohr, C., Nemitz, E., O'Dowd, C., Ovadnevaite, J., Pandis, S. N., Petäjä, T., Poulain, L., Saarikoski, S., Sellegri, K., Swietlicki, E., Tiitta, P., Worsnop, D. R., Baltensperger, U., and Prévôt, A. S. H.: Organic aerosol components derived from 25 AMS data sets across Europe using a consistent ME-2 based source apportionment approach, Atmospheric Chemistry and Physics, 14, 6159-6176, 2014.

Heikkinen, L., Äijälä, M., Riva, M., Luoma, K., Daellenbach, K. R., Aalto, J., Aalto, P., Aliaga, D., Aurela, M., Keskinen, H., Makkonen, U., Rantala, P., Kulmala, M., Petäjä, T., Worsnop, D., and Ehn, M.: Long-term sub-micrometer aerosol chemical composition in the boreal forest: inter- and intra-annual variability, Atmospheric Chemistry and Physics, 20, 3151-3180, 10.5194/acp-20-3151-2020, 2020.

Isokääntä, S., Kari, E., Buchholz, A., Hao, L., Schobesberger, S., Virtanen, A., and Mikkonen, S.: Comparison of dimension reduction techniques in the analysis of mass spectrometry data, Atmos. Meas. Tech., 13, 2995–3022, https://doi.org/10.5194/amt-13-2995-2020, 2020.

Stein, S. E., and Scott, D. R.: Optimization and testing of mass spectral library search algorithms for compound identification, Journal of the American Society for Mass Spectrometry, 5, 859-866, 10.1016/1044-0305(94)87009-8, 1994.

Ulbrich, I. M., Canagaratna, M. R., Zhang, Q., Worsnop, D. R., and Jimenez, J. L.: Interpretation of organic components from Positive Matrix Factorization of aerosol mass spectrometric data, Atmospheric Chemistry and Physics, 9, 2891-2918, 10.5194/acp-9-2891-2009, 2009.

Äijälä, M., Heikkinen, L., Fröhlich, R., Canonaco, F., Prévôt, A. S., Junninen, H., Petäjä, T., Kulmala, M., Worsnop, D., and Ehn, M.: Resolving anthropogenic aerosol pollution types–deconvolution and exploratory classification of pollution events, Atmospheric Chemistry and Physics, 17, 3165-3197, 2017.

Äijälä, M., Daellenbach, K. R., Canonaco, F., Heikkinen, L., Junninen, H., Petäjä, T., Kulmala, M., Prévôt, A. S., and Ehn, M.: Constructing a data-driven receptor model for organic and inorganic aerosol–a synthesis analysis of eight mass spectrometric data sets from a boreal forest site, Atmospheric Chemistry and Physics, 19, 3645-3672, 2019.